cognition/psychology/behaviour

processing fluency, source memory, familiarity, recollection, recognition memory, priming

**Author for correspondence:**
Tina S.-T. Huang
e-mail: shih-ting.huang.15@ucl.ac.uk

# Examining the relationship between processing fluency and memory for source information

## Tina S.-T. Huang and David R. Shanks

Division of Psychology and Language Sciences, University College London, 26 Bedford Way, London WC1H 0AP, UK

 TS-TH, 0000-0003-3693-1539; DRS, 0000-0002-4600-6323

Familiarity-based processes such as processing fluency can influence memory judgements in tests of item recognition. Many conventional accounts of source memory assume minimal influence of familiarity on source memory, but recent work has suggested that source memory judgements are affected when test stimuli are processed with greater fluency as a result of priming. The present experiments investigated the relationship between fluency and the accuracy of source memory decisions. Participants studied words presented with different source attributes. During test, they identified words that gradually clarified on screen through progressive demasking, made old/new and source memory judgements, and reported confidence ratings for those words. Response times (RTs) recorded from the item identification task formed the basis of a fluency measure, and identification RTs were compared across categories of item recognition, source accuracy and confidence. Identification RTs were faster in trials with correct retrieval of source information compared with trials for which source could not be accurately retrieved. These findings are consistent with the assumption that familiarity-based processes are related to source memory judgements.

## 1. Introduction

We often experience situations that require us to recall a certain piece of information and to be able to identify the origins of that information. For example, we might remember enjoying a particular dish, but we would also need to remember which restaurant we tried the dish in, in order to be able to have it again in the future. Source memory refers to memory for where, when or how a piece of information was acquired. The Source Monitoring Framework [1] conceptualizes source memory as involving a series of decision and evaluation processes, resulting

in varying degrees of precision and accuracy. As an individual continuously processes information from his or her environment, memory traces are tacitly formed which could later be reactivated to guide source attributions. In contrast with source memory, item memory refers to memory for a focal stimulus or event, but does not require its spatio-temporal context or associated features.

It is widely accepted that source memory tasks are more difficult than item memory tasks (e.g. [2,3]). Several factors are known to impact source memory accuracy. Poor attention during encoding can lead to incomplete revival of source information and impair source memory to a greater extent than item recognition [4]. Source memory accuracy can also be compromised when sources are highly similar in perceptual, semantic or conceptual qualities [5], and has been shown to be more vulnerable to ageing than item memory accuracy [6]. Although source memory is concerned with aspects that are often beyond our focal centre of attention at a given point in time, the ability to remember source information enables us to evaluate our knowledge and beliefs, and is, therefore, essential for day-to-day social interactions and for our subjective experience of autobiographical recollection. Failures of source memory underlie phenomena such as cryptomnesia [7] and confabulation [8]. Additionally, it has even been argued that most tests of recognition memory are essentially source memory tests [9]. Since all participants have encountered countless words and images outside of the experimental session, most recognition tests in effect require participants to discriminate study list stimuli from items encountered extra-experimentally, with perhaps the exception of some studies employing novel or unfamiliar stimuli (e.g. kaleidoscope fractals [10]).

In addition to item memory, source memory is often tested in recognition memory experiments which require participants to identify 'old' items from a study list and 'new' items which were not presented previously. When a test item is recognized as 'old', participants might simply *know* (K) that the item was previously encountered based on a sense of *familiarity* elicited by the item, or they might *remember* (R) experiencing the event of the item being presented based on their *recollection* of the episodic details associated with the item (we henceforth refer to this as the 'R/K procedure') [11]. Thus, source memory is commonly used as a proxy for recollection. Numerous models have been proposed to explain the roles of familiarity and recollection in recognition memory. According to dual-process models, familiarity and recollection are assumed to be two fundamentally distinct processes which contribute to recognition decisions and correspond, respectively, to the subjective experiences of knowing and remembering [12–15]. Despite some points of disagreement on areas such as the time course and neural substrates of familiarity and recollection, the general consensus among dual-process theorists is that familiarity is a faster process than recollection, and that the two processes operate independently at the time of retrieval [16].

Although closed-ended source memory questions may not always offer a comprehensive measure of recollection (for example, it is possible to recollect source information that has been neglected by the question, such as remembering that one had to sneeze when studying the word 'pocket'), it is a more objective measure compared with self-reports of familiarity and recollection such as the R/K procedure. Despite the close ties between source memory and recollection, Hicks, Marsh and Ritschel [17] found that correct perceptual source memory judgements are not necessarily accompanied by a greater number of R than K judgements. This implies that there may be some qualitative differences between source memory and generic recollection. Specifically, the high proportion of correct source responses that corresponded to K responses in Hicks *et al.*'s experiment [17] suggest that source judgements can successfully use partial memorial information which lacks clarity or detail.

Most dual-process accounts assume that familiarity is driven by a continuum of memory strength, whereas recollection arises from an all-or-none threshold process in which memory for episodic and source information occurs only in high-confidence instances of item retrieval, but not at all in other retrieval instances. Yonelinas [18] proposed that familiarity cannot be used for source judgements when two source attributes are of approximately equal familiarity (e.g. words spoken by either a male or a female experimenter), as is the case in most source monitoring experiments. Consistent with dual-process assumptions, Perfect *et al.* [19] showed that participants were usually able to make accurate source identifications for items with R responses but not for items with K responses. It has also been observed that both recollection and source memory performance can show impairments despite item recognition performance remaining intact in amnesic patients [20], patients with frontal lobe damage [21] and older participants [22].

On the other hand, single-system models of recognition memory interpret R and K response decisions as being driven by a unidimensional continuum of memory strength rather than by two discrete memory systems (e.g. [23–26]). A signal detection process is assumed by single-system models to underlie the evaluation of memory strength. Since the judgement criterion is expected to be higher for R than for

K, single-system models predict that an item receives an R response if its memory strength signal value is above the R criterion, whereas it receives a K response if its strength falls between the K and R criteria. Although dual-process models predict that any degree of recollection is only associated with the highest level of confidence, the single-system view is that some degree of recollection can be involved regardless of memory strength. Hence, familiarity can be interpreted as a weaker memory strength signal compared with recollection, but can nonetheless contribute to the recollective process.

Supporting the single-system view, studies have demonstrated that the recollection of source information can be assisted by vague, incomplete information such as whether a to-be-remembered item was pleasant or unpleasant, and that an above-chance proportion of K responses can be accompanied by accurate source judgements [27,28]. Through the use of state-trace analysis, R judgements were shown to be related to the 'old'/'new' recognition hit rate in a monotonic but nonlinear fashion, consistent with the signal detection process assumption of the single-system account, and the analysis also showed no evidence of R/K item recognition judgements being distinguishable by two separate latent variables [29]. Despite a greater number of neuroimaging and neuropsychological studies seemingly supporting the dual-process account, many also suggest that recollection (e.g. [30,31]), and even source memory [32], share similar or overlapping neural structures and processes with recollection. Conversely, a single-system model remains challenged by exceptional demonstrations of reversed associations [33] between familiarity (priming) and recollection, where retrieval intentionality produced opposite effects on the two measures (i.e. a stronger version of double dissociation) [34,35], although these results have not consistently been replicated [36].

Related to the single-system view, global matching models explain memory strength as being determined by the similarity between the retrieval cues and the contents stored in memory. Retrieval cues are holistically matched against all of the contents of memory, producing a single, summed or averaged [37,38] memory strength value that indexes the familiarity of the cues, that can then be compared with a response criterion to make a memory decision. Using global matching models, recent development by Osth *et al.* [39] described both source memory and item recognition as a global matching process, but in the case of source memory, the retrieval cues use additional source features. Critically, their models predict strong relationships between item recognition and source memory. While the list strength effect (i.e. strengthening some items impairs memory for other items [40]) is frequently observed in recall tests, Osth *et al.*'s Experiments 2 and 3 [39] found a lack of any such effect in both item recognition and source memory, which was consistent with their global matching models of source memory. The finding itself provided evidence against dual-process accounts (e.g. source of activation confusion (SAC) [41,42]; Norman & O'Reilly [43]) which assumed recollection to be impaired by list strength, and familiarity-based item recognition to be unaffected by it.

Attempting to reconcile single-system and dual-process approaches, Wixted & Mickes' [44] continuous dual-process model acknowledges the necessity of the signal detection assumption in single-system models, but noted its weakness in explaining memory content—particularly in the case of strong, high-confidence, familiarity-based recognition in the absence of contextual details ('butcher-on-the-bus phenomenon' [45]). Their continuous dual-process model extended the single-system signal detection model by adding recollection as an additional orthogonal, yet continuous, signal detection process. As the model assumes that confidence in recognition judgements is always predicated upon the sum of the two signals, it maintains the same predictions as the univariate signal detection model, but predicts separate mechanisms for R and K. Therefore, the model still accounts for high-confidence familiarity responses where the summative signal is high, but the recollection signal alone is low, or at least not high enough to pass the R criterion. Using a butcher-on-the-bus source memory paradigm, Tunney *et al.* [46] found an interaction between confidence estimates and R/K judgements, where source memory performance was above chance for high-confidence R *and* K trials. Tunney *et al.* proposed that, although neither traditional single-system nor dual-process models could explain their finding, it could be accommodated by Wixted & Mickes' [44] model if it additionally allows for familiarity and recollection to be partially correlated.

## 1.1. Fluency and priming effects on memory

Repetition priming, a phenomenon of implicit memory in which the fluency (i.e. speed and accuracy) of processing a stimulus is enhanced as a result of previous exposure, has been extensively studied in relation to recognition memory. Participants in Jacoby & Whitehouse's experiment [47] showed a greater tendency to rate primed items as 'old' regardless of whether they had previously encountered them during study. Using a similar repetition priming paradigm, Rajaram [48] investigated the impact

of fluency on remember/know judgements and found an increase in the proportion of K responses for primed versus unprimed trials, with no significant effects on the proportion of R responses. Although Rajaram's finding appears to support the dual-process prediction that fluency manipulations selectively affect familiarity, its generalizability has been criticized on methodological grounds for having a small sample size, potential type-2 error inflation and use of binary R/K categories as opposed to independent ratings [49–51].

Despite those limitations in Rajaram's study, several recent studies with higher power have successfully replicated the finding that presenting masked repetition primes immediately before test items induced a greater increase in K than R responses [52–54]. While the resulting single dissociations between priming and R/K in studies such as those by Jacoby & Whitehouse [47] and Rajaram [48] fall short of providing conclusive evidence for or against the single-system account, well-powered double dissociations have later been demonstrated. These provide a stronger case for distinct memory processes, for example, through cross-over effects of perceptual versus conceptual masked primes on K versus R judgements [53,54]. Further replication of such effects would be necessary to determine the extent to which they might be dependent upon particular experimental conditions (e.g. intermixing of prime types within the same block).

Apart from R/K judgements, source memory reports are another method commonly used to assess the conscious recollection of a prior episode, yet fewer experiments to date have investigated the role of fluency and priming in source memory. One example is Kelley *et al.* [55], in which participants studied a list consisting of auditorily and visually presented words, and during each trial in the test phase, they were asked to first identify rapidly presented old or new words, and then make a seen/heard/new judgement. Using participants' probability of identification as an indirect index of perceptual fluency, Kelley *et al.* demonstrated that correctly identified items were more likely than incorrectly identified items to be judged as previously seen, irrespective of whether the items were previously seen, heard or new.

Another example is Kurilla [56], which reported a series of experiments using a masked repetition priming paradigm. In Experiments 1A and 1B, participants were presented with primed and unprimed target and lure words, and were instructed to make either an old/new decision followed by a seen/heard source decision (1A) or a combined seen/heard/new decision (1B). Experiment 2 tested whether masked repetition priming may also affect source memory for different perceptual features within the same modality (i.e. two different font types). Results indicated that for primed items, participants were more likely to report that they had studied the words in the same font style that matched the font presented in the test phase, irrespective of whether the words were actually presented in the same font at study and test. Similar to Kelley *et al.*'s experiments [55], Kurilla's two experiments showed that priming increased the proportion of 'seen' responses but had no effect on the proportion of 'heard' responses, and Kurilla concluded that primed items have a greater tendency to be endorsed as being presented in the same sensory or perceptual form during study and test.

Although studies such as those by Kelley *et al.* [55] and Kurilla [56] imply that source memory judgements can be affected when processing fluency is manipulated through priming, they did not address whether fluency is related to source memory accuracy, nor did either of those studies directly measure variations in fluency. The continuous-identification with recognition (CID-R) paradigm [57] allows concurrent measures of priming, fluency and recognition to be obtained for every test item. In the CID-R test procedure, an old (target) or new (lure) item from the study list initially appears at a fragmented level. The item then gradually clarifies via a progressive demasking paradigm, and participants press a button immediately when they can identify the item, and then make a recognition response (e.g. old/new or confidence rating). Their identification response times (RTs) form the basis of a measure for priming and fluency. Therefore, the CID-R task additionally allows for a finer-grained measurement of variations in fluency than the perceptual identification task used by Kelley *et al.* [55].

Using a modified CID-R paradigm to examine the relationship between priming and R/K judgements, Berry *et al.*'s Experiment 3 [58] found that trials with R responses were associated with the fastest identification RTs, followed by K trials and 'new' trials. If fluency can contribute to source memory decisions, one implication of Berry *et al.*'s findings is that faster identification RTs on the CID-R task might occur for trials in which participants are able to correctly remember source information compared with trials on which source judgements were incorrect. The present experiments employed a test procedure combining CID-R and source memory confidence ratings in order to empirically examine this expectation. The experiments additionally investigated some of the ecologically relevant factors which may moderate the relationship between processing fluency and source memory: the depth of processing during encoding and temporal versus spatial source monitoring (Experiments 2a and 2b), and the number of competing contextual associations (contextual fan; Experiments 3a and 3b). The results are

initially analysed through analyses of variance, but later in the report, we also present an alternate analytic approach using linear mixed-effects modelling which enables the consideration of stimulus effects, as well as the examination of whether individual-trial identification is associated with source memory accuracy.

# 2. Experiment 1

The main purpose of Experiment 1 was to empirically investigate how fluency relates to source memory responses, in addition to the more prevalently used R/K or R/K/G responses. Another purpose of the experiment was to address the lack of research on memory involving multiple source attributes which are crossed at encoding and jointly retrieved [59]. Multidimensional source memory is ecologically important because episodic context is rarely constrained by a single source dimension in everyday life. To illustrate, we can often simultaneously retrieve multiple pieces of source information pertaining to an item which we own, such as where and when we acquired the item. Despite source memory being regarded as a multidimensional construct in the Source Monitoring Framework [1], the vast majority of source memory experiments have tested only one dimension of source information (e.g. colour) with two attributional variations (e.g. red/green).

One example of multidimensional source memory research is Meiser & Bröder [60]. In their first experiment, Meiser and Bröder crossed the font size (small/large) of study words with presentation location (upper/lower screen location) and demonstrated that participants' memory for one source dimension of an item was better when they correctly remembered the other source dimension than when they did not. This demonstration of stochastic dependence between memory for individual source dimensions suggests that remembering a source attribute on one dimension may facilitate or cue the retrieval of a source attribute on another dimension. A subsequent study [61] showed that, under perceptual encoding conditions, the proportion of items receiving correct source judgements on both dimensions was greater for R than K items, whereas the proportion of items receiving a correct source judgement on just one of the two dimensions was greater for K items than R items.

In the light of these findings, the present experiment investigated the relationship between identification RTs and source memory accuracy for font size and location. If the joint retrieval of features from two contextual dimensions depends exclusively on conscious recollection, it is unlikely that identification RTs on the CID-R task corresponding to items with correct responses on both source dimensions will differ significantly from those corresponding to correctly recognized items with incorrect responses on both dimensions.

In addition, as R judgements are previously reported to be associated with greater subjective confidence ratings and more accurate source memory on unidimensional source memory tests [62], the present study explored how R and K responses correspond to source memory confidence ratings collapsed across two contextual dimensions. Given the findings of previous experiments, we predicted that R responses would receive the highest source confidence ratings followed by K then G responses, and that the proportions of trials with zero, one or two sources correctly retrieved would vary as a function of R, K and G item ratings.

## 2.1. Method

### 2.1.1. Participants

Fifty University College London (UCL) students participated in the experiment for partial course credit or cash payment (£7.50). Two participants had to be discarded from data analyses for failure to follow task instructions, leaving an effective sample of 48, consistent with the sample size used in Meiser & Bröder [60], $M = 21.7$ years old, s.d. = 3.14; 38 females, 10 males. For this and all subsequent experiments, all participants spoke English fluently, reported normal or corrected-to-normal vision, and provided written consent for taking part in the experiment which was approved by the Ethics Committee of the UCL Department of Experimental Psychology.

### 2.1.2. Design and materials

The design of the study was adapted from the paradigms developed by Meiser & Bröder [60] and Stark & McClelland [57]. The source attributes and study phase procedures were nearly identical to those used by Meiser and Bröder. The test phase procedures mostly followed Stark and McClelland's CID-R procedure. However, instead of using old/new questions, participants were asked to make a remember/know/

guess/new judgement after identifying each test item, and were asked to provide confidence ratings for their memory of the two source dimensions associated with the test item if the item had received a 'remember' (R), 'know' (K) or 'guess' (G) judgement. The G response option was included in an effort to reduce guessing-related noise in K responses [63].

Experimental materials and instructions for this and all subsequent experiments were presented on a Dell PC monitor using Psychtoolbox 3 extensions [64–66] for MATLAB (Mathworks, Natwick, MA, USA). A total of 134 monosyllabic English nouns were selected from the MRC Psycholinguistic Database [67]. Each word had four to six letters, a Kučera–Francis Frequency score of 3–20, a Concreteness score of 400–670 and an Imageability score of 424–600. Sixty-four of these words served as targets, which were presented at both study and test, and 64 served as lures, which appeared only at test. Half of the remaining six words were used as primacy buffers which appeared at the beginning of the study phase and the other half were used as recency buffers which appeared at the end of the study phase. The buffer words did not appear on the test.

During the study phase, the appearance of each word could vary on two source dimensions: font size (small/large) and location on the screen (lower-centre/upper-centre location). The vertical axes of the words presented at the upper and lower locations were approximately 15 cm apart, with equal distance to the centre of the screen. The horizontal position of the words did not vary, as they were all presented at the centre of the horizontal axis of the display. The character heights for the small and large fonts were 0.7 cm (20 pt) and 1.8 cm (51 pt), respectively. The 64 target words were randomly selected, and their presentation was programmed such that a total of 16 words appeared in each of the four presentation formats (i.e. small font, lower location; large font, lower location; small font, upper location; and large font, upper location). Experimental instructions and all words appearing in the test phase (both targets and lures) appeared in a medium font size of approximately 1.2 cm (34 pt) character height. The Courier New font style was used for all stimuli and instructions in the experiment. A microphone headset was used for audio recording during the test phase. All frequentist statistical analyses were conducted using RStudio running R v. 3.3.1 [68] in this and all subsequent experiments of this paper, and accompanying Bayes factors were calculated using JASP [69] using the default priors provided in the software [70].

### 2.1.3. Procedure

Participants went through the procedure individually in a single session which lasted approximately 50 min. Before the study phase began, they were informed that words would appear on screen one at a time and were asked to 'try to remember as many of them as [they] can' and that their memories would be tested during the second part of the experiment. The instructions for the study phase did not make any reference to the varying appearance of the words, nor did the instructions provide any further detail on the nature of the memory test. The 64 target words were displayed on screen one at a time for 4 s with an interstimulus interval of 1 s.

After the end of the study phase, instructions describing the test phase procedures appeared on screen. A typical test trial began with a CID-R task which required participants to identify a word gradually presented on screen via progressive demasking (figure 1). In the task, a mask (i.e. a string of six hash symbols: ######) was displayed at the centre of the screen for 500 ms in medium font size, then a target word or a lure word was presented for 17 ms at the same location and in the same font size as the mask, followed by another presentation of the mask for 233 ms. The word was immediately presented again for 34 ms, and the mask was then displayed for 216 s. Thus, the 250 ms word–mask displayed block was repeated with the display duration of the word increased by 17 ms and the duration of the mask decreased by 17 ms in each subsequent repetition of the block until the mask duration was 0 (i.e. 14 blocks) or until participants made an identification response (see figure 1 for an illustration of the CID-R task and test questions).

To make an identification response, participants pressed the space bar as soon as they could identify the test word, and the RT to identifying the word was recorded. The word was replaced with the mask immediately after a participant's key press, and participants were instructed to read aloud the word which they had just identified. The microphone automatically recorded participants' identification responses which were verified offline by the experimenter. Participants then made a recognition judgement by clicking on the appropriate box whether they remembered, knew or guessed that the tested word had appeared in the study phase, or whether the word was new. The instructions for the four recognition response options were adapted from those used by Gardiner [71] and Eldridge et al. [63], and were described to the participant as follows:

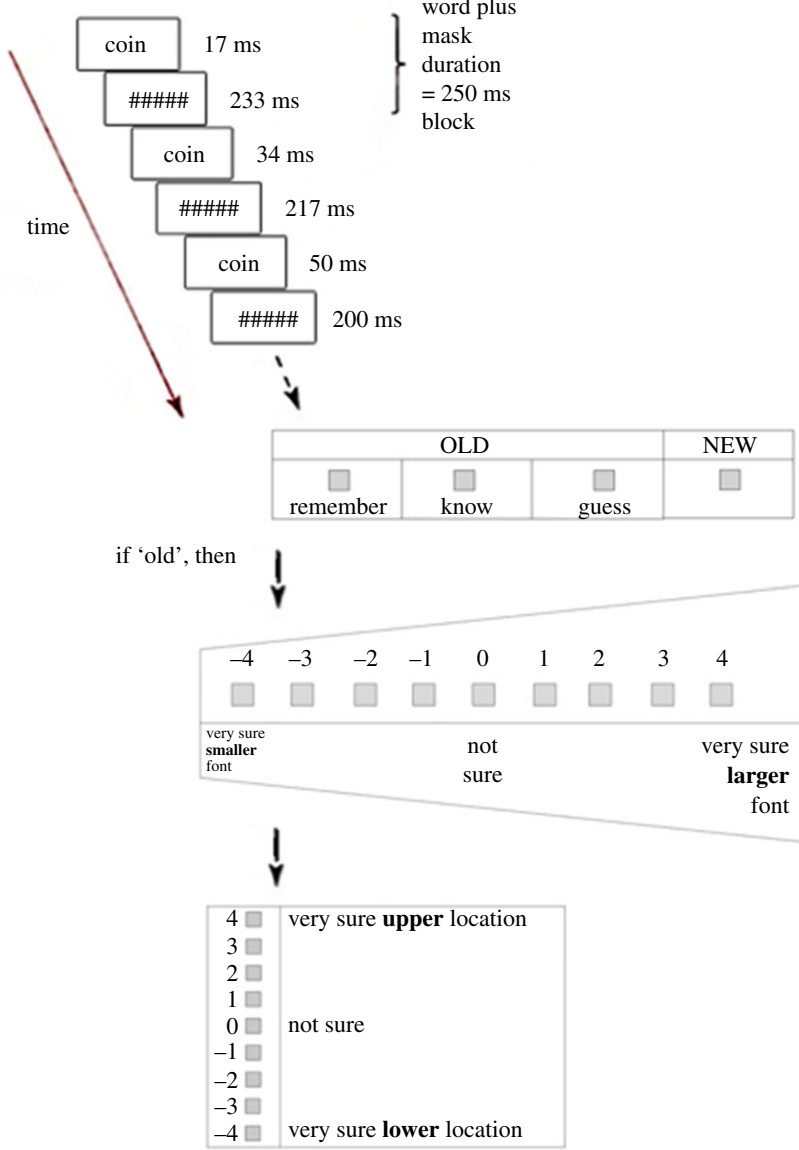

**Figure 1.** Diagram of the test procedures for Experiment 1.

If your recognition of the word is accompanied by a conscious recollection of your prior experience of the word during the study phase, select *REMEMBER*.

If you can recognise the word as having occurred in the study phase, but your recognition is not accompanied by a conscious revival of the event when the word was presented to you earlier, *KNOW*.

There might also be times when you do not remember the word, nor do you know it, but you might want to *GUESS* that it was one of the words you saw during the study phase.

If you think the word was NOT presented during the study phase, select *NEW*.

If participants made an R, K or G response, they proceeded to answer two source memory questions which asked them to rate on a nine-point confidence scale whether the test word had been studied in a larger or smaller font (−4 indicated 'very sure smaller font', 0 indicated 'not sure' and 4 indicated 'very sure larger font'), and whether it had appeared at the upper or lower screen location during the study phase (−4 indicated 'very sure lower location', 0 indicated 'not sure' and 4 indicated 'very sure upper location'). The order of presentation of the two source questions was counterbalanced between participants.

For trials for which participants reported the word being 'new', they proceeded to the next test trial without being presented with the source questions. On any particular trial, if participants were unable to identify the test word during the 14 presentation blocks (=3500 ms), a 'trial timeout' message appeared on screen, and participants directly proceeded to the next test trial. In total, there were 128 trials in the test phase per participant. All misidentified trials were excluded from subsequent data analyses.

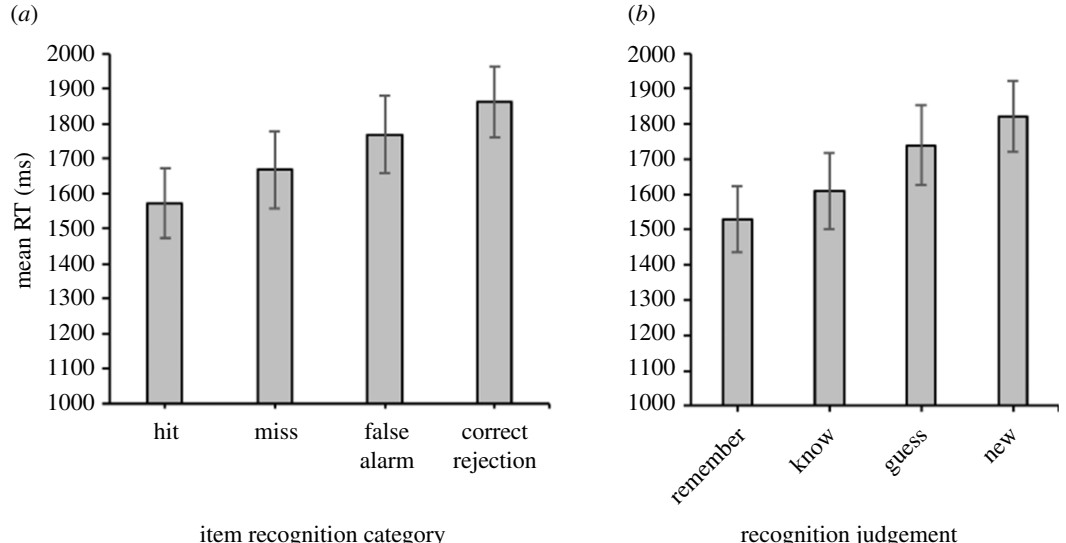

**Figure 2.** (*a*) Mean item identification RTs (ms) for hits, misses, false alarms and correct rejections in Experiment 1. (*b*) Mean identification RTs for item hit trials according to 'remember', 'know', 'guess' and 'new' response categories. Error bars indicate 95% confidence intervals of the mean.

## 2.2. Results

A total of 382 trials (6.22% of all trials) were excluded from the subsequent analyses due to misidentification of the test word or lack of an identification response. All 48 participants correctly identified the test word on at least 85% of their trials.

### 2.2.1. Recognition memory

Test trials receiving R, K and G were considered as 'old' responses. Across all valid test trials, the hit rate was 0.80 and the false alarm rate was 0.25.

Since the present experiment only collected RTs in the CID-R portion of the test phase, all references to identification RTs indicate RTs to the perceptual identification of the test item being presented at the beginning of each test trial. Figure 2*a* shows the mean identification RTs to hit, miss, false alarm and correct rejection trials.

A two-way repeated-measures ANOVA was conducted to test the relationship between subjective old/new judgements and actual old/new status of the items on identification RTs. There was a main effect of subjective old/new judgements on identification RTs, $F_{1,45} = 7.83$, $p = 0.0075$, $\eta_p^2 = 0.15$, $BF_{10} = 8.27$, as well as a main effect of actual status, $F_{1,45} = 56.99$, $p < 0.001$, $\eta_p^2 = 0.56$, $BF_{10} > 100$. This indicated faster identification RTs for old than new items, and for items judged as 'old' versus 'new', regardless of whether the items were actually presented in the study list. There was no significant interaction between old/new recognition judgements and the actual status of the items.

The mean RTs to R, K and G responses across item memory hit trials (figure 2*b*) were significantly different as determined by a one-way repeated-measures ANOVA, $F_{1.59,78.87} = 10.35$, $p < 0.001$, $\eta_p^2 = 0.18$, $BF_{10} = 5.85$. *Post hoc* t-tests revealed that identification RTs were faster for trials with R than K responses, $t_{43} = 2.75$, $p = 0.009$, $d = 0.42$, $BF_{10} = 4.45$, and K trials were associated, to a lesser degree, with faster identification RTs compared with G trials, $t_{41} = 2.55$, $p = 0.0431$, $d = 0.32$, $BF_{10} = 1.19$.

For 46 out of 48 participants, the mean identification RTs to old items were at least 10 ms faster than the mean RTs to new items. The mean identification RT for new items minus the mean identification RT for old items was calculated for each participant, and the overall difference indicated a significant priming effect, $M = 237$ ms, s.e.m. $= 36$ ms, $t_{47} = 7.87$, $p < 0.001$, $BF_{10} = 16.84$.

### 2.2.2. Source memory

Source judgements in item recognition hit trials were categorized as correct only if participants responded with an absolute confidence rating of 1 or more on the correct source. Participants' mean

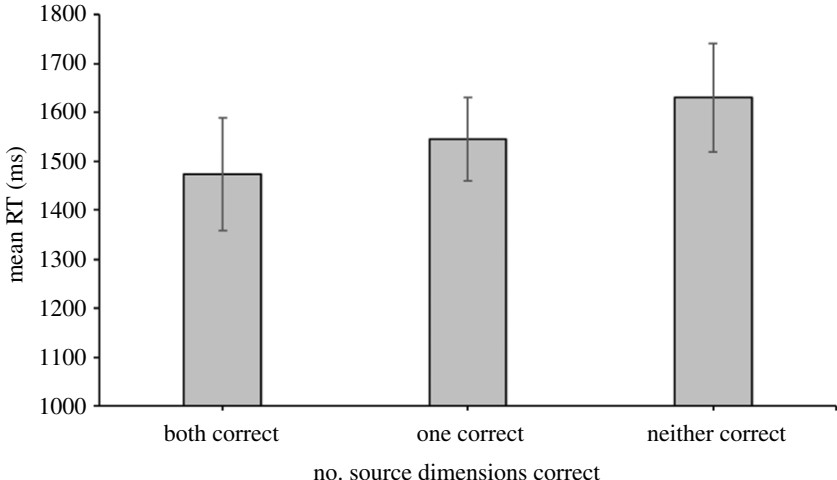

**Figure 3.** Mean identification RTs (ms) for recognition hit trials with correct source judgements on both, one or none of the source dimensions in Experiment 1. Error bars indicate 95% confidence intervals of the mean.

identification RT difference scores for correct versus incorrect source trials showed no significant difference between the size ($M = 99$ ms, s.d. $= 240$ ms) and location dimensions ($M = 114$ ms, s.d. $= 243$), $t_{47} = 0.59$, $p = 0.56$, $BF_{10} = 0.69$. Figure 3 shows the mean identification RTs according to the number of correct source identifications per trial. A one-way repeated-measures ANOVA[1] indicated a difference in identification RTs across trials with correct judgements on both source dimensions, trials with one correctly identified source, and trials with no correct source judgements, $F_{1.43,65.58} = 5.50$, $p = 0.0127$, $\eta_p^2 = 0.11$, $BF_{10} = 6.32$. *Post hoc* pairwise comparisons revealed that identification RTs for trials with both sources correct were faster than identification RTs for trials with no sources correct, $t_{46} = 2.67$, $p = 0.0319$, $d = 0.39$, $BF_{10} = 3.65$ and identification RTs for trials with one source correct were faster than identification RTs for trials with no sources correct, $t_{46} = 2.73$, $p = 0.0272$, $d = 0.40$, $BF_{10} = 4.19$. However, the difference in identification RTs between trials with two correctly identified sources versus trials with only one correct source was not significant, $t_{46} = 1.49$, $p = 0.4332$, $d = 0.22$, $BF_{10} = 0.44$.

A sign test was also conducted to examine the proportion of participants showing faster mean identification RTs for trials with two correct source judgements than for trials with no correct source judgements. In addition to the two participants excluded for having no trials with both sources correct, another participant was excluded for having tied mean identification RTs (i.e. a mean identification RT difference of less than 10 ms between trials with two correct source judgements and trials with two incorrect source judgements). Thirty participants showed faster mean RTs for trials in which both source judgements were correct compared with trials in which both source judgements were incorrect, and 15 participants showed the reverse pattern ($Z = 2.09$, $p = 0.0369$).

Further analyses were conducted to examine the relationship between item recognition responses, source accuracy and source confidence ratings. For R, K and G trials, table 1 shows response frequencies according to the number of correct source identifications per trial, as well as the mean total confidence ratings (sum of the absolute values of the confidence ratings for font size and location) per trial. A one-way repeated-measures ANOVA revealed a significant difference in confidence ratings per trial (collapsed across the two source dimensions) between trials receiving R, K or G responses, $F_{2,87} = 104.7$, $p < 0.001$, $\eta_p^2 = 0.71$, $BF_{10} = 58.25$. *Post hoc* pairwise comparisons with the Bonferroni correction demonstrated that R trials received higher overall source confidence ratings than K trials, $t_{44} = 6.27$, $p < 0.001$, $d = 0.93$, $BF_{10} = 11.83$, and K trials received higher source confidence than G trials, $t_{42} = 5.20$, $p < 0.001$, $d = 0.79$, $BF_{10} = 8.36$.

The relationship between item recognition responses and source memory accuracy was also analysed. Two participants were excluded from the analyses due to having no K or G responses. Across 46 participants, there was a significant and medium correlation between item-level R/K/G responses and source accuracy, Spearman's $\rho = 0.31$, $p < 0.025$. This indicated a tendency for more source dimensions to be correctly remembered when the corresponding recognition response was R, followed by K then G. Thirty-eight participants had positive correlations of $\rho \geq 0.10$, five had weak positive

---

[1]For this analysis, one participant was excluded for not having any trials with both sources correct.

**Table 1.** Source accuracy frequencies and confidence ratings for R, K and G trials according to the number of correct source judgements in Experiment 1. Source response frequencies represent the total number of trials with two, one or zero correct source judgements across all participants. The mean confidence ratings were obtained from the summed absolute confidence ratings for font size and location per trial (95% confidence interval of the mean is shown in parentheses).

| | source accuracy | | | |
|---|---|---|---|---|
| | both sources correct | one source correct | neither source correct | *M* summed confidence rating |
| remember | 509 | 510 | 412 | 4.14 (3.68, 4.60) |
| know | 122 | 199 | 243 | 2.38 (1.87, 2.90) |
| guess | 43 | 66 | 225 | 1.15 (0.72, 1.59) |

correlations of $0 \leq \rho < 0.10$ and the remaining three participants had weak negative correlations of $0 > \rho > -0.10$.

## 2.3. Discussion

The main objective of the present study was to investigate the connection between familiarity-related processes such as processing fluency and source memory, and in particular, their relationship to memory for multidimensional source information. A secondary objective was to explore how the accuracy of multidimensional source memory might be reflected in subjective confidence ratings. In pursuit of these objectives, trials with correct item recognition responses (item hits) were classified into three categories of source accuracy according to whether they had correct responses on both, one or neither of the source dimensions. Identification RTs were compared between these three categories of source accuracy in order to evaluate the relationship between fluency, source memory performance and subjective confidence ratings (as summed across the two source dimensions per trial).

The pattern of item recognition shown in the present study generally replicated the patterns reported previously in Berry *et al.* [58]. Overall, there was a trend for identification RTs to increase across hits, misses, false alarms and correct rejections. Identification RTs appeared to increase correspondingly across R, K and G trials, albeit the difference between the K and G trials may be considered weak in this experiment based on its Bayes factor [72]. A priming effect (i.e. faster mean RTs to target items than lures) was observed in the vast majority of participants. These results demonstrate that familiarity is linked to R decisions, and may challenge the interpretation of R responses as a process-pure index of recollection. Furthermore, identification RTs were predicted by both the old/new status of the items, as well as the subjective 'old'/'new' responses given by participants.

The positive correlation between R/K/G judgements and source accuracy in the present results was somewhat similar to those reported in previous cross-dimensional source memory studies which found above-chance source memory performance in K trials at the same time as an even greater frequency of accurate responses on both source dimensions in R trials [60,61]. Within R trials, we, however, did not find a numerically greater proportion of trials with two correctly identified source dimensions compared with the proportion with only one correctly identified dimension. Instead of finding a greater proportion of K judgements than R judgements in trials with one correctly retrieved source, the opposite pattern was shown in the present results. These differences may be due to the use of a one-step R/K/G/N question procedure here instead of the two-step procedure (i.e. old/new, then R/K) at test employed in previous experiments.

Even though the inclusion of a G response option has been demonstrated to reduce noise in K decisions by alleviating guessing-induced false alarm rates [63], it has been shown that K hit rates in one-step recognition tests remain lower than in two-step procedures, even after the addition of a G option [73]. Consequently, despite accounting for the number of G item hits, the proportion of K item hit trials in the present study is disproportionately smaller than the proportions reported in previous studies, which may have contributed to a floor effect. However, it would have been impractical to use the two-step recognition procedure, which would have rendered the test phase too lengthy. Apart from the aforementioned differences, the present findings largely agree with previous findings demonstrating a relationship between R/K/G judgements and multidimensional source accuracy and

a trend for confidence ratings to increase across R, K and G trials. This suggests that fluency corresponds to a subjective feeling of remembering the experience of an episodic event. However, the results do not yet rule out the possibility that correlations between the two measures (priming and source memory) may in fact be reflective of a single cause (e.g. attention) having similar effects on two independent processes (fluency and recollection).

Considering the debate about how the format of R and K instructions could affect patterns of memory judgements at test [48,74,75], it may also be of methodological interest to assess whether the present results could be replicated under different R/K instructions. For instance, the instructions for K responses in Experiment 1 excluded any references to the word 'familiar'. However, given that the experiential state of 'know' is known to colloquially connote higher item confidence, and the experiential state of 'familiar' is associated with retrieval of more information [76], using a 'familiar' instruction may result in better source accuracy for items classified as 'familiar', yet it is uncertain whether this might also change the items' corresponding identification RTs.

Of particular interest to the question of whether multidimensional source memory judgements are related to familiarity-based processes such as fluency, results from the present experiment indicated that identification RTs differed depending on source accuracy: trials with both sources correct tended to have faster identification RTs than trials with neither of the sources correct. This suggests that processing fluency can be associated with memory for more complete and accurate source information similar to how familiarity, when operationalized as K responses, can accompany partial source retrieval [59–61].

Despite obtaining slower identification RTs for trials with no correct source responses compared with trials with one and two correct source responses, the results revealed no significant RT difference between trials with both versus one of the source dimensions correct. Although this finding might appear to suggest that familiarity contributes to multidimensional source memory in a threshold rather than continuous manner, it is important to note the high probability of participants being able to correctly guess one of the two source dimensions, providing that a correct 'old' response was given earlier in a given test trial. Therefore, we do not believe this finding provides strong evidence for the dual-process account against the single-process account. Further research is needed to determine the precise phenomenological similarities and differences between partial versus complete retrieval of source information, and this may require the use of more than two attributes per source dimension in order to reduce the likelihood of correctly guessing one of the sources [6].

The present experiment presented two distinct source dimensions crossed at encoding, in an attempt to achieve a closer approximation of source monitoring in real-life situations. However, real-life source memories are undoubtedly more complex and involve a richer variety of features that characterize the circumstances under which information is encoded.

# 3. Experiment 2a

A primary finding from Experiment 1 is that accurate memory for source information is related to greater processing fluency as indexed by identification RTs on the CID-R task. Experiment 2 aimed to extend this finding to other encoding conditions and source modalities, by investigating the extent to which a levels of processing (LoP) manipulation at encoding might affect processing fluency and memory for temporal and spatial source information. Although items that are processed deeper and more elaborately at study are generally remembered better in explicit memory tests, Roediger *et al.* [77] argued that this LoP effect may vary depending on the perceptual, conceptual, explicit and implicit nature of the test. Numerous studies have failed to detect an effect of LoP on implicit perceptual tests such as perceptual priming (e.g. [3]) and word pair free-associations [78], despite showing such effects on explicit tests of recognition [13,79]. If deeper processing creates a stronger memory strength signal, then as the signal increases, one might assume a greater explicit/recognition memory advantage over priming/implicit memory performance.

In contrast with the aforementioned findings, Challis & Brodbeck [80] found an LoP effect on priming in a between-subjects word fragment completion task, whereas Roediger *et al.* [81] observed reversed LoP effects in both word fragment and word stem completion tasks involving pictorial stimuli, such that greater priming occurred in the graphaemic encoding condition relative to the pleasantness rating condition. Assuming that a given study has sufficient sensitivity and power to detect effects on priming, a dual-process account would readily explain the pattern of an LoP effect on recognition in the absence of any effects on priming, whereas a single-system account can explain the pattern of an LoP effect on priming with an LoP effect on explicit memory measures such as recognition.

LoP effects have also been demonstrated on explicit tests of associative memory for word pairs [82] and face–name pairs [83], but fewer studies have examined how LoP specifically affects source memory. In their Experiment 3, Glanzer *et al.* [84] found an LoP effect on item recognition and source memory, with deeper processing at encoding being linked to improvements for both item recognition and external source memory (male versus female voice). A study by Ragland *et al.* [85] demonstrated an LoP effect on an internal source monitoring task which instructed participants to identify at test whether a word was a target presented during the deep or shallow encoding condition, or was new, such that participants showed better internal source discrimination for semantically versus perceptually processed words. However, given the dissociations in general external and internal source monitoring performance (e.g. ageing; [86]), it is unclear whether Ragland *et al.*'s results [85] would generalize to external source monitoring tasks such as the one used in our Experiment 1.

Recent studies using R/K and R/K/G reports as an index of familiarity and recollection found that memory for the order [87] and sequence [88] of stimulus presentation could be retrieved accurately based on either familiarity alone or a combination of familiarity and recollection, although accurate source memory for other contextual dimensions was found to depend exclusively on recollection. Persson *et al.*'s experiments [88] were conducted in an immersive virtual environment such that each item in the study phase was presented in the context of one of six possible weather conditions (e.g. rainy weather was characterized by the use of visual and sound effects of raindrops in the background), and temporal source judgements involved six possible sequence position options. Their results showed that contextual source memory performance was above chance for R responses but not F responses, whereas performance for temporal source memory was above chance for both R and F. On the basis of those results, Persson *et al.* supported the dual-process account but proposed that the role of familiarity in temporal source memory is an exception, since memory strength at retrieval can reflect the amount of time elapsed since the presentation of that stimulus at encoding.

The aim of the present experiment was to investigate whether identification RTs to trials with correct source responses would be faster for identification RTs on incorrect source responses for temporal and spatial dimensions, and how LoP might affect these identification RT differences. Although each target word had a temporal placement (first/second half) and screen location (upper/lower) at study, the experiment was not designed to test multidimensional source memory as in Experiment 1, thus each test trial focused only on one of the two available dimensions. If deeper processing contributes to a stronger memory strength signal during encoding, we were interested to explore whether this could be reflected in implicit (processing fluency) as well as extraneous (temporal–spatial context) components of the memory. Based on the findings from previous experiments on associative memory [82,83], temporal source memory [87,88] and Experiment 1, we predicted that deeper processing at encoding would produce more accurate source memory responses, and that trials with correct source responses would have faster identification RTs, for both time and location, compared with trials with incorrect source responses.

## 3.1. Method

### 3.1.1. Participants

A total of 112 undergraduate students at UCL participated in the experiment as part of a laboratory class. Two participants were excluded from data analyses for failing to correctly identify more than 50% of target items at test, leaving an effective sample of $N = 110$; $M = 19.00$ years old, s.d. = 1.24. Half of all participants were randomly assigned to the shallow encoding condition of the experiment and the other half to the deep encoding condition, with 42 females and 13 males in each condition.

### 3.1.2. Design and materials

The experiment had level of processing (shallow versus deep) during encoding as a randomized between-subjects factor and source memory dimension (time versus location) as a within-subjects factor. Two orienting questions were used in each encoding condition, which randomly alternated between each encoding trial. Shallow orienting questions which emphasized orthographic aspects of the item included 'Does this word contain the letter "a"?' and 'Is this word exactly 5 letters long?'. Deep orienting questions which emphasized semantic aspects of the item included 'Is this word bigger or smaller than a shoebox?' and 'Is this word living or non-living?'. The experiment was not designed to examine participants' internal source monitoring for linking the specific orienting question to each item at test.

Rather, we decided to include two orienting questions per level of processing in order to ensure construct validity while preventing participant fatigue at encoding.

Unlike Experiment 1, which tested participants' memory for both source dimensions on each test trial, the present experiment only tested one dimension (i.e. time of presentation or screen location) per test trial, due to greater time constraints imposed as a result of the experiment being set within the students' class period. The selected source dimension was pre-allocated by the computer program to each test trial such that if a participant was able to progress to the memory judgement stage on all test trials, there would be an equal number of trials testing memory for time and for location. Item identification was immediately followed by a test for recognition *and* either temporal or spatial source memory (this contrasts with Experiment 1, in which memory for both dimensions was tested on each trial). In other words, in the interest of time and practicality, the R/K/G/N question appearing after the CID-R component in each test trial in Experiment 1 was omitted and replaced by a one-step question involving a single O/N judgement with the O subcategorized into a ratings scale consisting of six options based on source identification and confidence. Another difference from Experiment 1 is that the zero-confidence option (not sure) was no longer available. This was implemented to discourage potential conservative biases in responding.

All experimental materials and instructions were presented on a Dell PC monitor in the Courier New font style with a font size of 1.2 cm (34 pt). A total of 70 monosyllabic English nouns were selected from the MRC Psycholinguistic Database [67]. Each word had four or five letters, a Kučera–Francis Frequency score of 1–82, a Concreteness score of 487–648 and an Imageability score of 335–617. Thirty-two of these words served as targets, 32 served as test lures (which were also evenly and randomly assigned to each of the orienting questions), three words served as primacy buffers, and three served as recency buffers. Half of the words presented at study were five-letter words, contained the letter 'a', were living objects or were smaller than a shoebox. During the study phase, the vertical axes of the words presented at the upper and lower screen locations were approximately 15 cm apart and were equidistant from the centre of the screen.

### 3.1.3. Procedure

The experiment lasted approximately 15 min, and participants were tested in individual cubicles. Before the study phase began, they were informed that words would appear on screen one at a time along with questions, and were asked to answer those questions while trying to 'remember as many of the words as [they] can'. The instructions for the study phase did not make any reference to the varying appearance of the words, nor provide any further detail on the source memory aspects of the upcoming test. During the study phase, the 32 target words and six buffer words were displayed one at a time in black font for a total of 4 s each, and 500 ms after the onset of each word, one of the two orienting questions within each LoP condition was alternately selected per trial to be presented at the centre of the screen in blue font. Participants in the shallow LoP condition used the 'y' and 'n' keys to respond 'yes' or 'no', respectively, to the questions, and participants in the deep LoP condition used the keys '1' or '0' to respond 'bigger'/ 'living' or 'smaller'/'non-living', respectively. Once the responses to the orienting questions were recorded, the question then disappeared from the screen whereas the word remained on screen until 4 s had elapsed since the presentation onset of the word. The interstimulus interval between each word was 1 s. After participants had been presented with 19 words, the message 'You are now halfway through the study list. Press ⟨spacebar⟩ to continue' was displayed in a green font colour at the centre of a blank screen.

After participants had studied the second half of the study list, instructions explaining the test phase procedures were presented on screen. In total, there were 64 trials in the test phase per participant, each beginning with a CID-R task involving a target or lure word. The CID-R task was identical to the one used in Experiment 1 with two exceptions: the mask now consisted of five hash symbols instead of six, and participants were instructed to type the word they had identified rather than speak their answers aloud due to equipment constraints. If the participant could accurately identify the word on time, they were asked to make a single-step judgement of item recognition and temporal or spatial source memory. This differs from Experiment 1, in which memory for both source dimensions was tested on each trial. The six 'old' recognition options were grouped as a confidence scale with −3 indicating 'very sure lower location'/'very sure first half' and 3 indicating 'very sure upper location'/ 'very sure second half'. In place of 'very sure', lower confidence ratings were labelled 'probably' (for −2 and 2) and 'guess' (for −1 and 1). The 'new' option was located to the right of the 'old' options.

The source-O/N question was not presented on any test trial in which participants failed to correctly identify the word on time. If the participant was unable to identify the test word during the 14 presentation blocks (=3500 ms), a 'trial timeout' message appeared on screen, and the participant

directly proceeded to the next test trial. If the participant pressed the space bar on time but entered the wrong word, they saw an 'incorrect word entered' message before being directed to the next trial. All misidentified trials were excluded from subsequent data analyses.

## 3.2. Results

Across all participants, 335 trials (4.8% of all trials) were excluded from the subsequent analyses due to misidentification of the test word or lack of an identification response. All participants correctly identified the test word on at least 71.9% of their trials.

### 3.2.1. Recognition memory

The item recognition hit and false alarm rates were 0.85 and 0.20, respectively, across all valid trials. The item hit rate did not differ significantly between the deep ($M = 0.88$, s.d. = 0.12) and shallow ($M = 0.85$, s.d. = 0.12) conditions across participants, $t_{108} = 1.57$, $p = 0.1202$, $d = 0.30$, $BF_{10} = 0.50$, but there was a significant difference in the false alarm rate, $M_{(deep)} = 0.14$, s.d.$_{(deep)} = 0.10$, $M_{(shallow)} = 0.29$, s.d.$_{(shallow)} = 0.11$, $t_{108} = 4.18$, $p < 0.0001$, $d = 0.80$, $BF_{10} = 5.85$. Corrected recognition scores (i.e. Hits + Correct Rejections) were significantly higher for participants in the deep condition ($M = 0.87$, s.d. = 0.10) versus those who were in the shallow condition ($M = 0.78$, s.d. = 0.11), $t_{108} = 4.57$, $p < 0.0001$, $d = 0.87$, $BF_{10} = 7.21$.

There was a significant mean difference between identification RTs to new and old items ($M = 142$ ms, s.d. = 173) which indicated a reliable priming effect across participants, $t_{109} = 8.62$, $p < 0.0001$, $d = 0.82$, $BF_{10} = 25.52$. Across all participants, 89 provided evidence of priming with the mean identification RTs to old items being at least 10 ms faster than the mean RTs to new items, 18 showed RT differences in the opposite direction, and three had tied RTs (i.e. differences of 10 ms or less; $Z = 6.77$, $p < 0.0001$). Priming did not differ between the deep ($M = 127$ ms, s.d. = 168) and shallow ($M = 152$ ms, s.d. = 175) conditions, $t_{108} = 0.42$, $p = 0.6725$, $d = 0.14$, $BF_{10} = 1.52$.

### 3.2.2. Source memory

All analyses on source memory performance were carried out using data from item hit trials. A three-way mixed ANOVA was conducted to examine the effects of LoP, source dimension and source memory accuracy on identification RTs (figure 4a,b). The ANOVA found no main effect of LoP ($BF_{10} = 0.11$), nor were there any other significant main effects, with $BF_{10} < 1.00$ for all effects in this analysis. However, there was a significant three-way interaction, $F_{1,105} = 8.27$, $p = 0.005$, $\eta_p^w = 0.07$, $BF_{10} = 0.01$. Although the Bayesian results of this ANOVA showed a lack of an interaction effect (which is examined further in the Discussion), as our original experimental design and execution were based on frequentist analyses, we followed up this interaction with two separate two-way ANOVAs for each LoP condition. The shallow condition ANOVA revealed a significant interaction between source dimension and source accuracy, $F_{1,54} = 11.79$, $p = 0.0012$, $\eta_p^2 = 0.18$, $BF_{10} = 1.19$. Following up on the interaction in the shallow condition, tests of simple main effects found that the mean identification RTs were significantly longer when the temporal source dimension was incorrectly remembered than when it was correctly remembered, $F_{1,54} = 14.01$, $p < 0.001$, $\eta_p^2 = 0.22$, $BF_{10} = 4.44$, whereas no significant difference in RTs to correct versus incorrect source trials was observed when the location dimension was tested, $F_{1,54} = 1.24$, $p > 0.05$, $\eta_p^2 = 0.01$, $BF_{10} = 0.64$. On trials testing the temporal dimension in the shallow LoP condition, 33 participants showed slower mean identification RTs when the dimension was incorrectly remembered, 20 showed the opposite pattern and two had tied identification RTs with RT differences of 10 ms or less ($Z = 1.65$, $p = 0.0993$). There were no significant main effects of source dimension ($BF_{10} = 0.11$), source accuracy ($BF_{10} = 0.14$), nor an interaction ($BF_{10} = 0.02$) in the ANOVA for the deep condition.

Another three-way mixed ANOVA examined the effects of LoP, source dimension and source memory accuracy on confidence ratings (figure 4c,d). There was a significant main effect of source dimension on source confidence ratings, such that higher mean ratings were given on trials testing the temporal source dimension versus trials testing the location dimension, $F_{1,104} = 60.73$, $p < 0.0001$, $\eta_p^2 = 3.69$, $BF_{10} > 100$. There was also a significant main effect of source memory accuracy on confidence ratings, such that higher mean ratings were associated with correct source responses compared with incorrect source responses, $F_{1,106} = 65.60$, $p < 0.00001$, $\eta_p^2 = 3.82$, $BF_{10} > 100$.

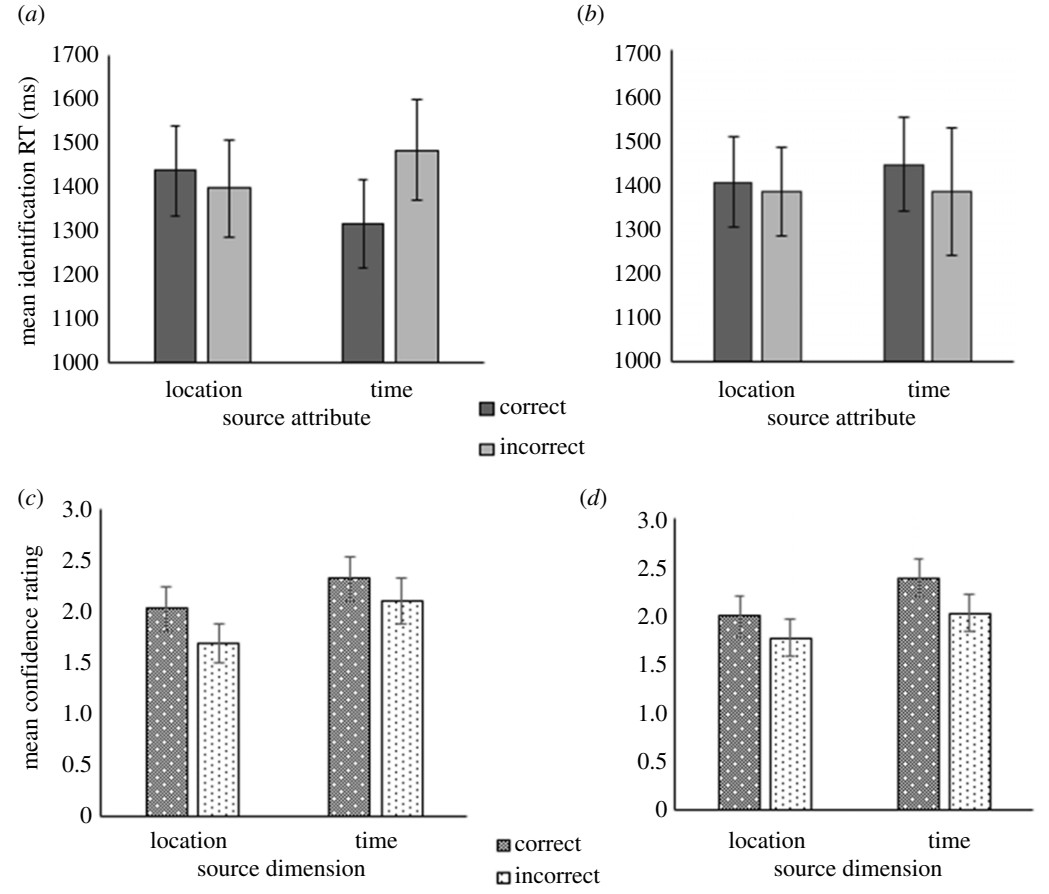

**Figure 4.** Effects of LoP, source dimension and source memory accuracy on identification RTs (*a,b*) and source confidence ratings (*c,d*) in the shallow (*a,c*) and deep (*b,d*) LoP conditions of Experiment 2a. Error bars indicate 95% confidence intervals of the mean.

## 3.3. Discussion

The item memory results from Experiment 2a were consistent with most of the recognition and priming results reported in previous LoP studies. Although deeper processing during encoding did not produce higher item recognition hit rates compared with shallow processing, it resulted in greater recognition accuracy as measured by the total number of hits and correct rejections. Priming at the item level was also unaffected by LoP, which corroborated previous findings employing perceptual identification tests (e.g. [13]). On a broader level, the source confidence results supported Experiment 1's finding that trials with correct source responses also tended to receive higher confidence ratings. Despite participants' higher confidence ratings on the temporal source dimension compared with the location source dimension, their performance on the source memory task did not indicate better memory for temporal attributes versus source attributes.

Within item hits, identification RTs were shown to be unrelated to LoP, with a Bayes factor providing moderately substantial support for this finding. As with prior findings obtaining an LoP effect on recognition in the absence of any effects on priming [13,79], the aforementioned result may be more easily explained by dual-process accounts than by single-system accounts of recognition memory. Analyses on the relationship between LoP, source memory accuracy and source dimension on identification RTs revealed an interaction between the three factors. Experiment 1's pattern of faster identification RTs associated with correct source trials compared with incorrect source trials was only obtained in the temporal source dimension within the shallow LoP group. However, the evidence for this may be presently speculative, as both the three-way interaction, as well as the interaction between source dimension and source accuracy within the shallow condition, yielded Bayes factors lower than 3. In particular, the Bayes factor associated with the interaction leaned in favour of a null effect, contrasting with the frequentist results. Such a contrast between Bayesian and frequentist results could arise in cases with a highly specific null hypothesis accompanied by a more diffuse alternative hypothesis and a neutral prior distribution (see the *Jeffreys–Lindley's paradox* [72,89,90]).

The lack of an LoP effect on source memory accuracy contradicted previous results on other associative forms of memory [82,83] and internal source monitoring [85]. This could be due to the possibility that the deep or elaborative encoding instructions used in those studies tended to promote more unitization (i.e. the encoding of an item with its surrounding contextual elements as a single, coherent unit of information), which can enhance associative memory [91] and source attribution accuracy [18]. For example, Cohn & Moscovitch [82] instructed participants in their deep associations condition to 'produce a sentence, aloud, that contained the two words, was meaningful, and maintained both the form (i.e., singular) and order as they appeared on the screen', and Troyer *et al.*'s [83] participants generated 'a definition or association for the name and then generated an activity for the face that was semantically related to the name'. Since Ragland *et al.*'s [85] source dimension of interest was the encoding instructions themselves (i.e. specifically whether items were processed shallowly or deeply during study), such information was highly salient at the time of encoding, and could have already been unitized with the items as part of the encoding process. By contrast, our deep encoding instructions did not direct any attention towards the items' screen location or time of presentation, and their focus on the semantic aspects of the items could have discouraged participants from unitizing items with their perceptual source information.

In the shallow condition, it was unexpected that the faster identification RTs to trials with accurate versus inaccurate source responses occurred in the temporal source dimension but not in the location dimension, given that Experiment 1 demonstrated faster identification RTs to trials with accurate versus inaccurate source responses in the visual source modality. A possible explanation for this interaction is Persson *et al.*'s proposal [88] that the contribution of familiarity to source memory is only exclusive to temporal forms of source memory. However, there are many fewer test trials on either source dimension compared with Experiment 1 (i.e. 16 versus 64 trials, respectively), and thus some participants in Experiment 2a might not have sufficient time to familiarize themselves with the test format. As the test duration was far shorter, the smaller amount of data collected per participant could have contributed more noise to the results of this experiment.

# 4. Experiment 2b

The aim of this experiment was to use a larger number of study and test trials per participant to replicate Experiment 2a's finding that identification RTs were faster for trials with correct time judgements compared with trials with incorrect time judgements in the shallow LoP condition, whereas identification RTs did not differ between trials with correct and incorrect location source judgements. Therefore, this experiment did not include a deep LoP condition.

## 4.1. Method

### 4.1.1. Participants

Based on the effect size ($d_z = 0.70$) corresponding to the standardized identification RT difference scores between trials with correct versus incorrect temporal source responses in the shallow LoP condition of Experiment 2a, a power analysis using G*Power (v. 3.1.9.2; [92]) indicated that a minimum of 18 participants would be required to detect a significant difference ($\alpha = 0.05$) at 0.80 power. Twenty-one UCL students participated in the experiment for partial course credit or cash payment (£5.00). One participant was removed from data analyses for not following task instructions, leaving an effective sample of $N = 20$; $M = 24.45$ years old, s.d. = 6.07, 13 females, and seven males.

### 4.1.2. Design and materials

The within-subjects factor of interest was source memory dimension (time versus location). This experiment had the same design as that of Experiment 2a, except that orienting questions were omitted during encoding, and the stimulus set included 64 target and 128 test items (twice as many than the number of items used in Experiment 2a).

A total of 134 monosyllabic English nouns were selected from the MRC Psycholinguistic Database [67]. These included the 70 nouns from Experiment 2a and additional nouns with similar properties, and thus all nouns had four or five letters, a Kučera–Francis frequency score of 1–312, Concreteness scores of 487–670 and Imageability scores of 335–643. For each participant, each of the nouns was randomly assigned to be one of the 64 targets, one of the 64 lures, one of the three primacy buffers or one of the three recency buffers. All instructions and stimuli were presented in the same font styles,

font sizes and format as in Experiment 2a. A microphone was used to record participants' word identification answers at test, which were spoken aloud (see Experiment 1) rather than typed.

### 4.1.3. Procedure

Participants took part in the experiment individually in a single session which lasted approximately 40 min. As in Experiments 1 and 2a, the study phase procedure involved intentional encoding of the word items and incidental encoding of the source attributes. The same presentation durations and interstimulus intervals were used as in Experiment 1.

After the end of the study phase, further instructions explaining the test procedures appeared on the screen. The CID-R task was the same as the version used in Experiment 2a, except that participants were asked to identify words aloud after pressing the space bar instead of typing their answers. The recognition and source memory questions were also identical to the version used in Experiment 2a.

## 4.2. Results

Across all participants, 181 trials (7.1% of all trials) were excluded from the subsequent analyses due to misidentification of the test word or lack of an identification response. Each participant correctly identified the test word on at least 68.8% of their trials.

### 4.2.1. Recognition memory

The item recognition hit and false alarm rates were 0.79 and 0.25, respectively, across all valid trials. The overall difference between the mean identification RTs to new ($M = 1755$ ms, s.d. $= 529$) and old ($M = 1592$ ms, s.d. $= 496$) items indicated a significant priming effect across participants, $t_{19} = 6.45$, $p < 0.0001$, $d = 0.32$, $BF_{10} = 8.65$, with 18 participants showing evidence of priming with the mean identification RTs to old items being at least 10 ms faster than the mean RTs to new items, and two participants showing RT differences in the opposite direction.

### 4.2.2. Source memory

All analyses on source memory performance were carried out using data from item hit trials. The mean identification RTs to trials with correctly and incorrectly responded source questions on time and location are shown in figure 5. Since our sample size was planned based on Experiment 2a's effect of source accuracy on identification RTs on the temporal dimension (rather than the interaction effect), the present source memory results focused on $t$-tests.[2] Consistent with the results from Experiment 2a, identification RTs were significantly faster on trials where participants correctly answered whether the item had appeared in the first or second half of the study phase, compared with when the presentation time was incorrectly answered, $t_{19} = 2.52$, $p = 0.02$, $d = 0.56$, $BF_{10} = 2.79$. For 15 participants, the mean identification RTs to trials with correct temporal source responses were at least 10 ms faster than to trials with incorrect temporal responses, four participants showed the reverse pattern, and one had tied identification RTs. As in Experiment 2a, there was no significant identification RT difference between trials with correct location responses versus trials with incorrect location responses $t_{19} = 0.60$, $p > 0.05$, $d = 0.14$, $BF_{10} = 0.27$.

## 4.3. Discussion

Using a larger number of study and test trials per participant, Experiment 2b replicated the pattern of findings in the shallow LoP condition of Experiment 2a. On the temporal source trials, mean identification RTs to source-correct trials were faster than to source-incorrect trials, but the same pattern was not observed on the location source trials. However, with a Bayes Factor slightly below 3, the evidence for the source accuracy effect on identification RTs in the temporal source dimension is less substantial compared with that obtained in Experiment 2a. Taken together, the results of Experiments 2a and 2b are generally consistent with those of Persson *et al.* [88]. According to Persson *et al.*'s proposal, the presence of an identification RT difference between correct and incorrect temporal

---

[2]An unplanned two-way ANOVA indicated a significant interaction effect of source dimension and source memory accuracy on identification RTs, $F_{1,19} = 6.23$, $p = 0.02$, $\eta_p^2 = 0.06$ concurring with Experiment 2a.

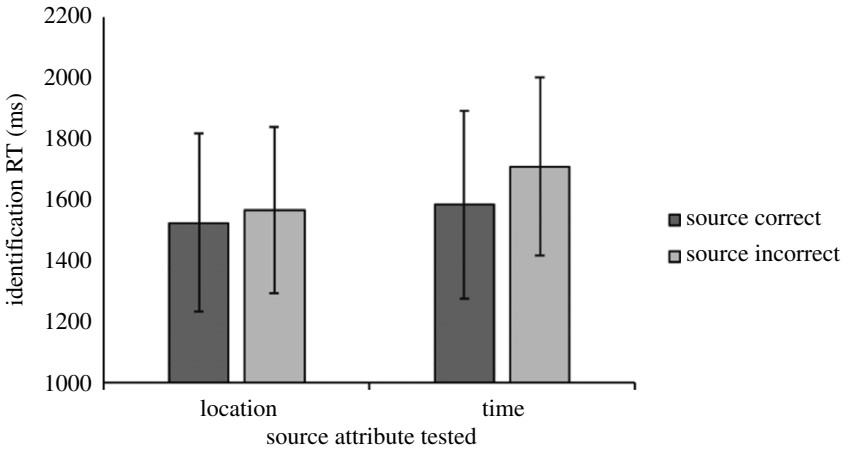

**Figure 5.** Mean identification RTs according to source memory accuracy and tested source dimension in Experiment 2b. Error bars indicate 95% confidence intervals of the mean.

source trials in Experiments 2a and 2b supports the contribution of familiarity (and fluency) in temporal source judgements, whereas the absence of identification RT differences on location trials indicates a lack of familiarity to visuospatial source judgements. However, our findings could not provide definitive evidence in favour of this conclusion, especially since identification RTs were shown to be faster to trials with correct versus incorrect source judgements on location and font size in Experiment 1.

An alternative explanation for the pattern of results in Experiments 2a and 2b could be provided by the source of activation confusion (SAC) model of memory by Reder *et al.* [93]. According to the SAC model, a memory trace is used to encode the fact that a particular item was studied in a particular experimental context during the study phase. When the item is presented at test, activation will spread to the relevant item and source information stored in memory. It is easier to retrieve the encoding context when there are fewer competing contextual associations (i.e. low contextual fan conditions), as the source information has higher distinctiveness and a higher amount of the source activation would be directed towards memory for its associated encoding event. Context retrieval would be more difficult in high-contextual fan conditions because there is greater associative interference, and the memory trace linking an item with its associated source attribute would become weaker as it is saturated by the greater number of other items also associated with the same source attribute. Supporting this model, Reder *et al.* [93] found that perceptual match effects (i.e. better memory for items that are presented with similar physical attributes at study and test) are enhanced for words presented in font styles not shared with other words.

In our Experiments 2a and 2b, the location source dimension could be considered as having a high fan since there were only two possible screen locations, and each location was shared with half of the study items, whereas the temporal dimension could potentially have had a much lower fan since each word had occupied a unique sequential position in time. If contextual fan could affect memory for item and source information in the manner suggested by the SAC, it is possible that the higher contextual fan in the location dimension could have attenuated the relationship between processing fluency and source judgements.

# 5. Experiment 3a

The results from Experiments 2a and 2b suggested faster identification RTs to trials with correct versus incorrect temporal source responses, and no difference in identification RTs between trials with correct and incorrect location responses. In those experiments, each word (which acted as the cue for source memory) appeared only once, and thus each cue was associated with only one source variant. It is important to reiterate that Reder *et al.*'s test [93] of the SAC model manipulated the number of source memory cues (i.e. words) associated with a single source (i.e. font style), and thus our conceptualization of 'fan' here is in the reverse direction of their manipulation. However, by doing a backwards manipulation of fan through increasing the number of source variants in the present experiment (with the number of words remaining the same), each source variant would effectively

achieve a lower fan. This is because each source variant would now be uniquely associated with a smaller number words, when compared with being associated with an entire half of the study list words.

Thus, the objective of Experiment 3a was to test the possibility that the relationship between processing fluency and source memory accuracy can be affected by the contextual fan of the source information associated with studied items. For this purpose, we manipulated the number of attributional variations within two visuospatial source dimensions: screen location and font colour (i.e. 2 versus 64 total variations within each of these source dimensions). Based on the predictions of the SAC, we expected a greater identification RT difference between source-correct and source-incorrect trials when the tested source dimension comprised more (low fan) variations in source attributes at study, in comparison with a source dimension with fewer source attribute variations (high fan).

## 5.1. Method

### 5.1.1. Participants and design

An *a priori* power analysis using G*Power (v. 3.1.9.2; [92]) indicated that a minimum of 34 participants would be required to detect a significant difference ($\alpha = 0.05$; two-tailed) of a medium effect size ($d_z = 0.50$) at 0.80 power. In total, 40 volunteers were recruited via the UCL Psychology Subject Pool to participate in exchange for £5.00. The design had contextual fan (low versus high) as the main within-subjects factor of interest. This was achieved through two conceptually equivalent between-subject groups. Both groups had one of the two available source dimensions presented with high-contextual fan and their other source dimension presented with low contextual fan: in the *low–location–fan/high–colour–fan* (LLF/HCF) group ($n = 20$; $M = 25.10$ years old, s.d. = 5.09; 14 females, six males), words appeared at multiple locations on the top or bottom of the screen in either a vermilion or cerulean font colour (high–colour–fan) at study, and the *low–colour–fan/high–location–fan* (LCF/HLF) group ($n = 20$; $M = 24.10$ years old, s.d. = 3.39; 14 females, six males) was presented with words displayed in a variety of blue–green and red–orange font colours at either the top-centre or bottom-centre of the screen (high location fan).

### 5.1.2. Materials

The present experiment used the same set of 134 monosyllabic English nouns as the one used in Experiment 2b. All stimuli and instructions were presented in a 34-point Courier New font as in Experiments 2a and 2b. The spectrum of font colours of the LCF/HLF group were generated using RGB Color Gradient Maker [94], with the 32 colours of the blue–green gradient ranging from vivid cobalt blue (#004ED2) to moderate spring green (#37A86B) and the 32 red–orange colours ranging from moderate crimson (#AA062E) to vivid orange (#D66C00). The two gradients were of approximately equal saturation and luminosity. In the LLF/HCF group, the vertical axes of the words presented at the upper and lower locations were all approximately 15 cm apart, but each of the top-location and bottom-location words was assigned to be presented in one of the 32 horizontal axis points which were equidistant across the width of the screen. The 64 total colours were randomly assigned to the 64 positions. The two font colours used in the LLF/HCF group were moderate cerulean (#1A79A0) and moderate vermilion (#BF3717) which were selected from the respective midpoints of the blue–green and red–orange gradients. A microphone was used to record participants' word identification answers at test.

### 5.1.3. Procedure

The study phase instructions given to participants at the beginning of the experiment were identical to the version used in Experiment 2b. During the study phase, the three primacy buffers, 64 target items and three recency buffers were each displayed on screen one at a time for 4 s, and between each word, a blank screen with a fixation cross at the centre was displayed for 1 s. For both contextual fan groups, primacy and recency buffers were randomly selected to appear at the top-centre or bottom-centre of the screen in either strong cerulean or vivid vermilion. Once the study phase ended, participants received instructions on the CID-R task and source memory questions which they would complete in the test phase. The test phase procedures were identical to those in Experiments 2a and 2b, the test items in the CID-R task were presented in a black font colour at the centre of the screen and the source memory options for font colour were displayed in the same layout as the temporal source options in those experiments, with the

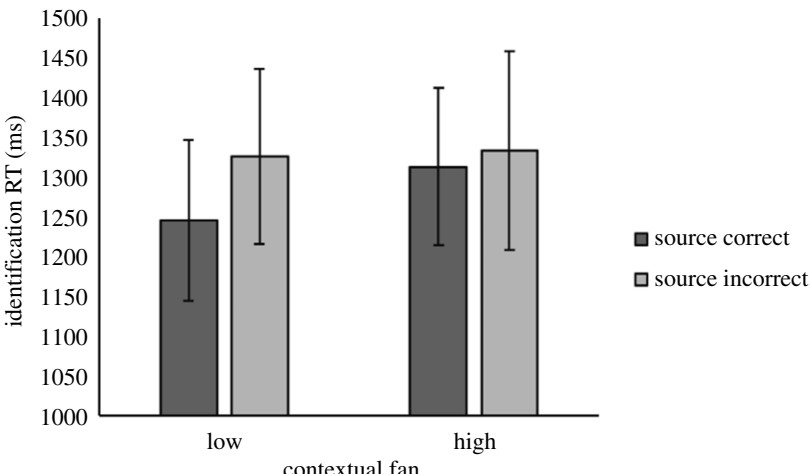

**Figure 6.** Mean identification RTs according to source memory accuracy in Experiment 3a. Error bars indicate 95% confidence intervals of the mean.

font of the text labels such as 'very sure blue/green'/'very sure red/orange' displayed in the same full spectrum of their respective colours as used in the study phase stimuli.

## 5.2. Results

Across all participants, 182 trials (3.56% of all trials) were excluded from the subsequent analyses due to misidentification of the test word or lack of an identification response. Each participant correctly identified the test word on at least 81.25% of their trials.

### 5.2.1. Recognition memory

The item recognition hit and false alarm rates were 0.84 and 0.28, respectively, across all valid trials. The overall difference between the mean identification RTs to new ($M = 1480$ ms, s.d. $= 328$) and old ($M = 1395$ ms, s.d. $= 355$) items indicated a significant priming effect across participants, $t_{39} = 4.23$, $p < 0.001$, $d = 0.67$, $BF_{10} = 5.21$, with 30 participants showing evidence of priming with the mean identification RTs to old items being at least 10 ms faster than mean RTs to new items, eight participants showing RT differences in the opposite direction, and two participants with tied mean identification RTs for old and new items.

### 5.2.2. Source memory

For all source memory analyses, low–location–fan and low–colour–fan trials were collapsed across participants as low-contextual-fan trials, and high–location–fan and high–colour–fan trials were collapsed as high-contextual-fan trials. The data analyses were not broken down by colour or location as we were interested in visual source modality as a whole, and the study was not designed with the sample size required for making such comparisons. A repeated-measures ANOVA was conducted to examine the effects of context fan and source accuracy on identification RTs (figure 6). The ANOVA indicated a significant main effect of source accuracy, $F_{1,37} = 4.66$, $p = 0.0372$, $\eta_p^2 = 0.11$, $BF_{10} = 0.98$, but neither the context fan effect, $F_{1,37} = 1.60$, $p = 0.2144$, $\eta_p^2 = 0.04$, $BF_{10} = 0.31$, nor the interaction, $F_{1,37} = 0.33$, $p = 0.57$, $\eta_p^2 = 0.01$, $BF_{10} = 0.09$, were significant. Twenty-nine participants had faster mean identification RTs to source-correct trials than to source-incorrect trials, nine showed the reverse pattern and two had tied mean RTs (i.e. with mean RT differences of less than 10 ms).

## 5.3. Discussion

Consistent with Experiments 1 and 2, identification RTs were faster for trials with correct source responses compared with trials with incorrect source responses. However, this source accuracy effect on identification RTs may be interpreted as anecdotal at best, due to its small Bayes factor. Contrary to our expectations, there was no significant effect of contextual fan. However, this may have been due to an artefact related to the within-participants manipulation of contextual fan. It is possible that

the contextual fan information could have been summed across word colour and locations whenever any degree of unitization between the item and its two source dimensions had occurred during encoding, thus resulting in equivalent net contextual fan across multiple trials.

# 6. Experiment 3b

Although lower contextual fan did not correspond to faster identification RTs in Experiment 3a, the results were not conclusive due to the possibility that any unitization between item, location and colour during encoding could have weakened the within-subjects contextual fan manipulation. To eliminate this possibility, Experiment 3b aimed to replicate the results of Experiment 3a by varying only one source dimension at the time of encoding rather than two, and by manipulating contextual fan within participants in two separate blocks.

## 6.1. Method

### 6.1.1. Participants and design

Following the *a priori* power and minimal sample size calculation from Experiment 3a, 44 volunteers participated in the experiment in exchange for partial course credit or £7.50 ($M = 22.55$ years old, s.d. $= 3.55$; 30 females, 14 males). Each participant was randomly assigned to either the location or the colour version of the task, and there were equal numbers of participants for each version. The experiment had contextual fan as the within-participants factor of interest. In the colour version of the task, this was achieved by presenting words in different shades of blue–green and red–orange in the study phase of the low-fan block, and in one shade of blue–green and red–orange in the high-fan block. In the location version of the task, the low-fan block involved words being presented in various top and bottom locations of the screen, and at only the top-centre location or the bottom-centre screen location in the high-fan block.

### 6.1.2. Materials and procedure

A total of 204 monosyllabic English nouns were selected from the MRC Psycholinguistic Database [67]. The words had four or five letters, a Kučera–Francis frequency score of 1–213, Concreteness scores of 406–646 and Imageability scores of 431–647. For each participant, each of the nouns was randomly assigned to appear in either the first or the second block of the experiment, acting as one of the 48 targets presented at study, one of the 48 lures at test, one of the three primacy buffers or one of the three recency buffers. All instructions and stimuli were presented in the same font styles, font sizes and format as in Experiment 3a.

The blue–green and red–orange colour spectra were generated using the same procedure as in Experiment 3a, but there were 24 instead of 32 shades in each spectrum. The screen locations of the words were generated with the vertical axis location of the word being randomly sampled from 24 equidistant points across the width of the screen, and the horizontal axis location fixed at approximately 15 cm from the screen centre for all top and bottom words. A microphone was used to record participants' spoken word identification answers at test.

The experimental session consisted of two study-test blocks: one high-fan block and one low-fan block, with the order of the blocks counterbalanced across participants. Before starting the experiment, the instructions given to participants were generally the same as in Experiment 3a, although participants were additionally informed about the two-block structure of the session, and that the two blocks had no relation with each other (i.e. they can forget about all the words from the first block once they finished it, as the second block would not test them on those words). In the low-fan block, participants studied words presented in one of 24 variations of blue–green/red–orange shades (top/bottom locations), whereas words were presented in only one blue–green/red–orange shade (top-centre/bottom-centre location) in the high-fan group. The CID-R task and question format of the test phase were also identical to that of Experiment 3a. After finishing the test phase of the first block, participants completed a 2 min word search puzzle as a filler activity before starting the second block. The study and test phases of the second block were conducted in the same procedural format as the first block.

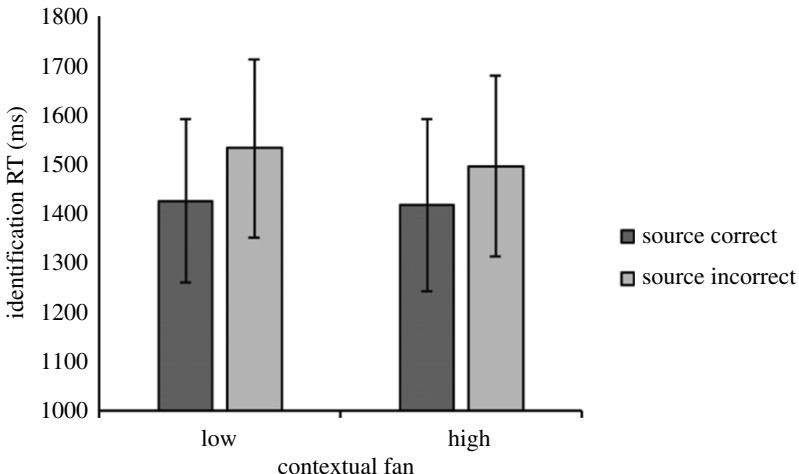

**Figure 7.** Mean identification RTs according to source memory accuracy in Experiment 3b. Error bars indicate 95% confidence intervals of the mean.

## 6.2. Results

Across all participants, 324 trials (3.84% of all trials) were excluded from the subsequent analyses due to misidentification of the test word or lack of an identification response. Each participant correctly identified the test word on at least 70.83% of their trials.

### 6.2.1. Recognition memory

Across all valid trials, item recognition hit and false alarm rates were 0.77 and 0.28, respectively. The overall difference between the mean identification RTs to new ($M = 958$ ms, s.d. = 387) and old ($M = 863$ ms, s.d. = 383) test items indicated a significant priming effect across participants, $t_{43} = 8.78$, $p < 0.00001$, $d = 1.32$, $BF_{10} = 17.5$, with 39 participants showing evidence of priming with the mean identification RTs to old items being at least 10 ms faster than the mean RTs to new items, two participants showing RT differences in the opposite direction and three participants with tied mean identification RTs for old and new items.

### 6.2.2. Source memory

A repeated-measures ANOVA was conducted to examine the effects of context fan and source accuracy on identification RTs (figure 7). The ANOVA indicated a significant main effect of source accuracy, $F_{1,42} = 8.69$, $p = 0.0072$, $\eta_p^2 = 0.17$, $BF_{10} = 1.79$, but neither the context fan effect, $F_{1,42} = 0.65$, $p = 0.4240$, $\eta_p^2 = 0.02$, $BF_{10} = 0.21$, nor the interaction, $F_{1,42} = 0.68$, $p = 0.4145$, $\eta_p^2 = 0.02$, $BF_{10} = 0.15$, were significant. Thirty-one participants showed faster mean identification RTs to source-correct trials than to source-incorrect trials, 10 showed the reverse pattern and three had tied mean RTs (i.e. with mean RT differences of less than 10 ms).

## 6.3. Discussion

In line with the previous experiments in this article, trials with correct source responses tended to have faster item identification RTs than trials with incorrect source responses. However, as in Experiment 3a, the evidence supporting the source accuracy effect on identification RTs should be considered anecdotal, due to their Bayes factors being less than 3. Replicating the findings from Experiment 3a, the current results show no evidence that contextual fan directly affects identification RTs or moderates the relationship between identification RTs and source memory accuracy. Although Reder *et al.* [93] found that low fan encoding conditions can further enhance perceptual match effects, it is possible that, as a factor on its own, contextual fan might not have as much of a direct influence on item–source recognition. In Buchler *et al.*'s study [95] on memory for paired associates, their high fan manipulation adversely affected the retrieval of associations but not items. Since identification RTs were made at the item level in Experiments 3a and 3b, it is possible that the SAC's predictions did not apply here.

It would also warrant further investigation to determine the extent that the 64 locations or colours were, in reality, perceived as 64 functionally distinct source attributes by participants. Although much care was put into selecting as diverse a range of locations and colours as possible, Miller [96] has suggested that people are able to make accurate absolute identifications for an approximate maximum of five to seven equally spaced stimuli along a perceptual continuum. Experiments 3a and 3b would then only have had 10–14 effective source variants in the low-fan conditions (i.e. five to seven perceivable variations in each of the upper *and* lower, or blue–green *and* red–orange, sources) versus two source variants in the high-fan conditions, as opposed to 64 versus two, and could thus have rendered the fan manipulations considerably weaker.

Another explanation for the lack of a fan effect on identification RTs or source accuracy could be due to the difference in the direction of fan manipulations of the source attributes (colour/location) and source memory cues (word items) used in this study, when compared with Reder *et al.* [93]. In our Experiments 3a and 3b, experiment-wide contextual fan was manipulated by arranging greater or fewer source attribute variations while keeping the number of word items constant. By contrast, Reder *et al.* [93] manipulated contextual fan by varying the number of words items associated with each individual font style. Although both types of manipulation ultimately alter contextual fan in terms of the experiment-wide source-to-cue ratio, there have been no studies directly comparing the effects of these two approaches, to the best of our knowledge.

# 7. Meta-analysis and linear mixed-effects analysis

The results across Experiments 1–3b suggested that processing fluency (identification RTs) are related to source memory accuracy under certain conditions. Experiment 1 showed faster identification RTs to trials with both source dimensions (font size and screen location) correctly identified versus trials with no correctly identified source dimensions. Experiment 2a found faster identification RTs to source-correct trials than source-incorrect trials under the shallow encoding condition, but only when the source attributes were temporal, and no such identification RT differences were shown when the source attributes were visuospatial (i.e. screen location). These results were replicated in Experiment 2b. However, visuospatial source attributes were used again in Experiments 3a and 3b (by collapsing the font colour and screen location dimensions in the analyses), and the results of both showed faster identification RTs to source-correct trials than source-incorrect trials.

Using the *metafor* package [97] in R, a meta-analysis with a random-effects model was conducted to examine the overall relationship between identification RTs and source memory accuracy. Across the aforementioned instances in Experiments 1–3b where source memory accuracy effects on identification RTs were found, the meta-analytic effect size was $d_z = 0.46$, 95% CI = [0.26, 0.66], a medium-sized effect. The heterogeneity between the experiments was non-significant, $Q(4) = 2.50$, $p = 0.64$.

Also collapsing the data from Experiments 1–3b, the relationship between identification RTs and source accuracy was examined at an individual-trial level with linear mixed-effects analyses, using the *lme4* [98] and *afex* [99] packages in R. First, we determined the optimal random-effects structure by using restricted maximum-likelihood estimation to fit a maximal linear mixed-effects model. The fixed effect factors of the model included: identification RTs as the primary outcome variable, the explanatory variable of source accuracy as the primary fixed effect of interest, as well as fixed effect terms for tested source dimension (visual versus temporal), Experiment (1–3b), LoP (deep versus shallow, with shallow processing assumed in Experiments 1, 3a and 3b) and the interaction of all the fixed effects terms [100]. Participant and stimuli (presented word items) were included as random factors, and thus we included in this maximal model random intercepts and slopes for all fixed effects by both participants and stimuli. If the model resulted in a singular fit, we removed the random effect slopes and/or intercepts which explained the lowest estimated amount of variance, until the model converged with a non-singular fit. The resulting optimal random-effects structure included random intercepts only for stimuli and participants. Random slopes for source accuracy, LoP, source dimension and experiment were dropped, due to their inclusion resulting in singular fits and lower Akaike's information criterion (AIC) values, as well as near-zero variance explained by their inclusion.

The final full model included source accuracy, tested source dimension, experiment, LoP and their interactions as fixed terms, and only the intercepts of stimuli and participant as the random terms. The results of this model, shown in table 2, indicated that individual-trial identification RTs differed by source dimension and experiment. Most importantly, individual-trial identification RTs were also strongly related to source accuracy, consistent with the analyses of variance reported in the previous sections for each experiment. The effect of the source dimension should be interpreted cautiously, as

**Table 2.** Results of a linear mixed-effects regression analysing effects on individual-trial identification RTs, with data collapsed across Experiments 1–3b. $p$-values for fixed effects calculated using Satterthwaite approximations. Model equation: identificationRT ~ SourceAccuracy × LoP × SourceDimension × Experiment + (1 | Stimulus) + (1 | Participant).

| | ANOVA: fixed effects | | |
| --- | --- | --- | --- |
| | d.f. | F | $p$-value |
| source accuracy | (1, 10 487.77) | 9.87 | 0.002 |
| LoP | (1, 491.75) | 0.37 | 0.545 |
| source dimension | (1, 10 226.64) | 5.82 | 0.016 |
| experiment | (4, 326.17) | 5.23 | <0.001 |
| source accuracy × LoP | (1, 10 395.53) | 9.60 | 0.002 |
| source accuracy × source dimension | (1, 10 475.69) | 0.83 | 0.361 |
| LoP × source dimension | (1, 10 301.29) | 2.52 | 0.112 |
| source accuracy × experiment | (4, 10 496.25) | 0.67 | 0.611 |
| source dimension × experiment | (1, 10 336.36) | 3.85 | 0.050 |
| source accuracy × LoP × source dimension | (1, 10 393.84) | 8.64 | 0.003 |
| source accuracy × source dimension × experiment | (1, 10 468.80) | 1.81 | 0.178 |
| | **random effects** | | |
| | **variance** | **s.d.** | |
| stimulus | 0.01 | 0.09 | |
| participant | 0.12 | 0.34 | |
| | **model fit** | | |
| | **AIC** | **BIC** | |
| | 17 035 | 17 188 | |
| | **marginal** | **conditional** | |
| $R^2$ | 0.02 | 0.34 | |

the number of trials testing the visual source dimension vastly outnumbered the temporal test trials. Given the procedural and sample variations between the experiments, it was not unexpected that identification RTs varied significantly between them. Single-trial level identification RTs did not significantly differ between LoP levels, although LoP appeared to interact with some of the other factors, but note that shallow processing trials vastly outnumbered deep trials. However, given the model's low marginal $R^2$, the above results should be taken with caution.

Secondly, in a binary/logistic linear mixed-effects model (table 3), we evaluated whether source accuracy at the individual-trial level could be predicted by individual-trial identification RTs. We first determined the optimal random-effects structure using the same procedure as in the previous linear mixed-effects model. In this analysis, source memory accuracy was the primary outcome variable and identification RT was the primary explanatory effect of interest. During this optimization process for the random-effects structure, source dimension, LoP and experiment were fixed factors, and for all of these, we also included random slopes and/or intercepts by participants and stimuli. The final model included identification RTs, source dimension, experiment and LoP as the model's fixed effect terms, with stimuli as a random factor and random intercepts and slopes for identification RT by participants. As illustrated in table 2, results revealed identification RTs as the only factor significantly predicting source accuracy ($p < 0.001$), although again this should be interpreted tentatively, given the model's low marginal $R^2$ and high AIC and BIC values.

# 8. General discussion

The current experiments demonstrate the relevance of perceptual processing fluency for source memory performance, supported by analyses of variance for each experiment, and corroborated by an inter-

**Table 3.** Results of a binary/logistic mixed-effects model analysing effects on individual-trial source memory accuracy, with data collapsed across Experiments 1–3b. $p$-values for fixed effects calculated using Laplace approximations of maximum likelihood. Model equation: SourceAccuracy ~ identificationRT × LoP × SourceDimension × Experiment + (1 | Stimulus) + (identificationRT || Participant).

| | analysis of deviance (Type II Wald $\chi^2$): fixed effects | | |
| --- | --- | --- | --- |
| | $\chi^2$ | d.f. | $p$-value |
| identification RT | 27.11 | 1 | <0.0001 |
| LoP | 0.03 | 1 | 0.87 |
| source dimension | 0.36 | 1 | 0.55 |
| experiment | 6.93 | 4 | 0.14 |
| identification RT × LoP | 6.34 | 1 | 0.01 |
| identification RT × source dimension | 10.63 | 1 | 0.001 |
| LoP × source dimension | 0.54 | 1 | 0.45 |
| identification RT × experiment | 8.59 | 4 | 0.07 |
| source dimension × experiment | 0.06 | 1 | 0.80 |
| identification RT × LoP × source dimension | 4.69 | 1 | 0.03 |
| identification RT × source dimension × experiment | 0.01 | 1 | 0.91 |
| | **random effects** | | |
| | **variance** | **s.d.** | |
| participant (intercept) | 0.22 | 0.47 | |
| participant (slope: identification RT) | 0.005 | 0.07 | |
| stimulus (intercept) | 0.02 | 0.14 | |
| | **model fit** | | |
| | **AIC** | **BIC** | |
| | 13 936 | 14 089 | |
| | **marginal** | **conditional** | |
| $R^2$ | 0.01 | 0.08 | |

experimental meta-analysis and by linear mixed-effects models at the individual-trial level. They reveal a pattern of association largely consistent with Kelley *et al.*'s results [55], which showed a dependence between perceptual identification at test with source modality judgements. It is also consistent with Kurilla's findings [56] that participants' tendency to report that an item had been studied in the same source format is influenced by artificially manipulating perceptual processing fluency. The present results additionally suggest that fluency is also linked to the accuracy of the source judgements. Furthermore, our participants had no access to the original source attributes at test, as all test items were presented in a neutral colour, font or location (unlike in Kelley *et al.*'s and Kurilla's studies where participants judged whether or not the source format of the test item matched its original source at study). Consequently, our participants would have had no access to familiarity via the perceptual match between the source attribute(s) of study and test items. This would have made the test even more dependent on recollection, yet the identification RT results imply that recollective processes were not the only contributors to test performance. It is unknown the extent to which participants were able to access their stored representations of the item *and* source information during the stimulus demasking phase of the CID-R task at test, and this could be an avenue for future research.

The relationship between item-level identification RTs and source accuracy and confidence challenges versions of dual-system models that specifically assume complete independence between the bases of implicit and explicit memory (e.g. priming and recognition, respectively [101]), given that recollection and the retrieval of source information are conventionally considered to be even more reliant on explicit processes than item recognition. This pattern is compatible, in contrast, with single-system models, although it may not necessarily contradict dual-system models which allow for implicit and explicit memory to be correlated (i.e. the MS2 model [58]). For example, when attention fluctuates

during encoding, some items may accordingly be better encoded than others, resulting in both faster identification RTs and more accurate source retrieval for those items at test. Thus, dual-system models which include a free correlation parameter between implicit and explicit memory strengths would allow for fluency and source memory performance to be partially related even if the two measures rely on separate systems. A zero-correlation is implied by dual-system models that assume complete independence of implicit and explicit bases, whereas a correlation value of 1 is implied by single-system models (i.e. both bases of memory strength signals can be represented as a single base).

The primary purpose of our experiments was an empirical exploration of how perceptual fluency relates to source memory, along with some of the factors which may moderate this relationship. The application of formal computational models (which we have undertaken elsewhere [58]) was beyond the scope and design of the study. In Berry et al.'s study [58] on fluency and recognition memory, the best fitting correlation parameter estimate was found to be 0.93 after applying the MS2 model to data across their three experiments. This near-maximal correlation suggested that empirically, the MS2 model's performance was nearly identical to that of the single-system model. Future studies might directly manipulate attention or distinctiveness during encoding in order to examine their influences on fluency, source accuracy and the correlation between them. Although it is uncertain to what extent such a high correlation parameter is similarly applicable to our specific results reported here, subsequent work may benefit from additional computational investigation comparing the ability of single- and dual-/multiple-systems models to fit the data, especially given the absence of dissociations between priming and source memory.

Independently of our present study, recent experiments by Lange et al. [102] demonstrated, also via a modified CID-R task, that both source confidence and accuracy were associated with faster identification RTs on the task. This relationship was enhanced particularly when the overall memory strength increased (via shortening the study list; Experiment 2). Lange et al. were able to show that a single-system model outperformed a multiple-systems model in reproducing the qualitative patterns of association between source memory and priming as found in their experimental data. Quantitatively, the single-system model also provided a superior fit in their Experiments 1 and 2, but not in their Experiments 3a and 3b. The latter two experiments showed a weaker association between priming and recognition memory, but their findings could still be explained by the MS2 model. Critically, the authors demonstrated that a multiple-systems model with a low correlation parameter between priming, recognition memory and source memory (MS1) provided a poorer fit for their results. Lange et al. further argued that, although the MS2 model with its higher correlation parameter estimate could, in effect, fit the priming and source memory data in Experiments 1 and 2, it only did so through mimicry of the single-system model and overfitting.

Consequently, Lange et al. [102] concluded that the single-system model provided the most parsimonious interpretation of their findings and should, on the whole, be preferred over an interrelated, multiple-systems model such as the MS2 model. At present, there are no existing reversed associations specifically between priming and source memory accuracy documented in the literature. The plausibility of finding such a reversed association may be a worthwhile avenue for future experiments, and should it be demonstrated empirically, it would provide a fruitful opportunity for further model comparisons and refinement.

Additionally, by measuring identification RTs and memory ratings in separate test phases, Lange et al. ([102]; Experiments 3a and 3b) addressed the possibility that this effect of fluency on item and source recognition confidence ratings might be a consequence specific to the interleaved structure of the CID-R task itself (which was also a limitation of our present experiments). As items are identified more quickly, participants might be more inclined to attribute fluency to the prior exposure of the item at study as a function of the memory judgements being immediately preceded by the identification [13]. In other words, even if performance on the CID task and the memory judgements are based on two separate systems, performance could be correlated simply due to the temporal proximity of the two tasks, which would still be consistent with dual-system models which allow for correlations between implicit and explicit bases of memory (e.g. the MS2 model; [58]). The results of Lange et al.'s Experiments 3a and 3b demonstrate that the relationship between identification RTs, source memory accuracy and source confidence persisted despite the separation of the CID and memory judgement components at test. This suggests that any influence of shared or sequential effects across items at retrieval (e.g. attention) is minimal. Therefore, one might also expect that the correlation parameter between the processes associated with priming and recognition/source memory would be similarly high in other paradigms that do not involve the trial-by-trial combination of a perceptual identification test immediately followed by a source memory judgement [103]. However, in such designs, it remains possible that the correlation between priming and recognition/source memory tasks is, to some extent, produced by the influence of any correlations at encoding (e.g. as a result of attentional fluctuation).

In addition to list length, as examined by Lange *et al.*, our experiments contribute to a better understanding of other ecologically relevant moderators of source memory. Experiments 2a and 2b showed an LoP effect on item recognition, as well as the presence of a fluency and source accuracy association for shallowly (but not deeply) processed items, and for visual rather than temporal source modality judgements. Experiments 3a and 3b showed no effect of contextual fan on either fluency or source accuracy. There are, nonetheless, some aspects of these secondary findings that are less consistent with the single-system model. Specifically, in Experiments 2a and 2b, the main effect of LoP was absent in identification RTs, despite affecting recognition (i.e. false alarms and correct rejections). This dissociation appears to be more consistent with the conventional interpretation of the lack of LoP effects on priming as evidence in favour of dual-process or multiple-systems accounts [79,80,104].

On the basis of the definition for source memory, having access to recollective details is commonly assumed to be crucial to determining the success or failure of source information retrieval (e.g. [18,19,105]). Nonetheless, the present findings are generally in line with the view that familiarity-related processes and other types of information can also support source memory (e.g. [1,59]). Although there is still ongoing research on the circumstances that would enable or promote the contribution of familiarity to source memory, Yonelinas [106] proposed this contribution can occur, especially when item and source information are encoded as a single unit. More recent studies have demonstrated an important role of item–source unitization in moderating familiarity's contribution to source memory (e.g. [41]). For their study phase, Diana *et al.* instructed participants in the high-unitization condition to imagine items as being in their corresponding background colours, and participants in the low-unitization condition to imagine items associated with background-coloured objects (e.g. if the background is red, associate item with a red stop sign). Based on converging behavioural and event-related potential (ERP) results, the study suggested that the unitization instructions increase familiarity's contribution to source memory.

It is possible that the items and source attributes in the present experiments were unitized to some degree at the time of encoding. Yet, it is also possible that very little unitization had occurred, given that most of our item–source associations would have been rather arbitrary (e.g. the word 'torch' presented in a green font colour), and that the presentation duration might not have been sufficient to allow for participants to generate unitized images themselves. According to the Levels of Unitization Framework [107], there can be lower (e.g. 'this torch was found in a green field') versus higher (e.g. 'this torch emits green light') degrees of unitization. Despite the former being a unitized image, it still consists of an arbitrary association between two separate entities, whereas the latter forms a single entity in its own right. Whether or not unitization is necessary in order to observe a relationship between fluency and source memory still needs to be further investigated, as well as the level of unitization that would have been required for this relationship to occur.

To conclude, the present findings established fluency as an important contributor to memory for source information at least on some dimensions. The exact nature of the relationship between fluency and source memory awaits additional study, as does the contribution of fluency to source monitoring in more ecological settings, but the present findings are among the first to reveal directly that familiarity-based processes are linked to source memory accuracy. Given the vital part source information plays in our social interactions and episodic remembering, research should continue to elucidate the mechanisms underlying source memory.

Ethics. The study was reviewed and approved by the Ethics Committee of the University College London Department of Experimental Psychology (no. EP/2015/5).

Data accessibility. All data and materials have been made publicly available via the Dryad Digital Repository: https://doi.org/10.5061/dryad.jq2bvq85t [108].

Authors' contributions. T.S.-T.H. and D.R.S. conceptualized and designed the study. T.S.-T.H. collected and analysed the data, and interpreted the results. T.S.-T.H. and D.R.S. drafted the manuscript. All authors gave final approval for publication.

Competing interests. We declare we have no competing interests.

Funding. There are no funders or financial support to report for this study.

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
