## [Peer Review File · Royal Society Open Science]

Review History

RSOS-190430.R0 (Original submission)

Review form: Reviewer 1

Is the manuscript scientifically sound in its present form?

Yes

Are the interpretations and conclusions justified by the results?

Yes

Is the language acceptable?

Yes

Do you have any ethical concerns with this paper?

No

Have you any concerns about statistical analyses in this paper?

No

Recommendation?

Accept with minor revision (please list in comments)

Comments to the Author(s)

This paper describes a series of behavioural experiments testing the relationship between fluency, as measured by identification times in a perceptual priming task, and memory for various types of source, using the CID-R paradigm previously employed by the authors' group. While the conclusions remain unclear, and it is a shame that the authors did not apply their published computational model, the experiments are well-conducted and powered, so will hopefully provide constraints on future theories and models.

1. The main finding is that correct source judgments are associated with faster identification times than incorrect judgments, and the (implicit) aim of the authors is to claim that this supports single-process models in which priming and source memory emerge from the same memory signal, contrary to the conventional view that implicit and explicit (certainly episodic) memory reflect different memory processes/systems. While I think it is good to question conventional wisdom, I am generally more convinced by dissociations than associations. There are numerous behavioural and neuroscientific dissociations between priming and episodic memory, and while I accept that many of these have faults, and that all single dissociations can be trivially explained by a single-process model, there are a few key findings (such as the reversed association of Richardson-Klavehn et al, 1999) that I think refute single-process models. By contrast, there are many trivial reasons for associations to emerge, even from multi-process models. For example, "random" fluctuations in attention from trial-to-trial during the study phase are likely to affect processes that underlie both fluency and recollection. In other words, correlations between two measures (priming and source memory) could reflect a single cause (attention) having similar effects on two, independent processes (fluency and recollection). This was on my mind throughout reading the paper, so it was gratifying to see the authors acknowledge this in the General Discussion. However, this point could have been made earlier, and the language could be modified to reflect this in several places; for example in the concluding paragraph, the authors say their study is "the first to reveal directly that familiarity-based processes influence source memory accuracy", but they have no direct evidence that familiarity *influences* (in causal sense) source memory, only that there are conditions under which the two are correlated (possibly because of a cause having parallel effects on two processes). Likewise for earlier statements like: "These results demonstrate that familiarity can contribute to R decisions, and contradict the interpretation of R responses as a process-pure index of recollection" – they do not demonstrate this directly.

2. In the absence of dissociations, a better way to compare single- vs dual-process theories is to fit formal models to the data, and ask whether the extra complexity of dual-processes models is justified by the data (i.e., an argument based on parsimony). The authors have done this elegantly in previous work (as they describe here in General Discussion) – but was there a reason why they didn't compare models in their ability (e.g., in terms of AIC/BIC) to fit the present data? Also, it's not clear to me why the authors focus here on their prior argument that the parameter that captures the correlation between the processes associated with priming and recognition memory was previously high (.93). I don't think this is as good evidence for a single-process model as formal measures of (penalized) model fit (e.g., AIC/BIC or cross-validation) and tests of model recovery/mimicry – if a dual-process model with high correlation between processes were still favoured by the data, then it would not matter if this correlation were high (perhaps some of the remaining $(1-.93)^2$ variance reflects important distinct variance in the data). More importantly, perhaps this parameter would be lower if fit to the present data, and perhaps it would be lower still in paradigms other than the CID-R paradigm where the trial-by-trial combination of a perceptual identification test immediately followed by a source memory judgment is likely to maximise the influence of factors that have similar effects on multiple memory processes (e.g., random fluctuations in attention during test trials, as well as study trials).

3. I would not insist on this, but would it be worth fitting linear mixed effects models (eg using “lmer” in R) to single-trial data? Not only might this give higher power and the ability to model stimulus effects, but it might allow tests of whether individual trial identification times “predict” (using a logistic/binomial linking function) individual trial source accuracy, over and above other factors?

Minor points

4. Does the statement: “Since all participants have encountered countless words and images outside of the experimental session...” in the Introduction apply to meaningless stimuli used in many recognition memory studies to address this issue, like the fractals used by Voss & Paller? I appreciate that participants might still use pre-experimental knowledge to try to interpret such stimuli, so the distinction is not clear-cut, but this might be worth mentioning.

5. Do the criticisms of Rajaram’s study in terms of a “small sample size” and “potential type-2 error inflation” apply to the several replications, such as Woollams et al (2008) and Taylor et al (2012, 2013) (Refs below). Personally, I think the evidence is quite convincing that masked repetition primes immediately before test items increase K judgments more than R judgments. I accept that this is only a single dissociation, so not conclusive evidence for a single-process models. Furthermore, I accept the final criticism regarding exclusive versus parallel procedures for R/K judgments (of Higham & Vokey). Nonetheless, there have been reports of a double dissociation, whereby masked conceptual primes affect R but not K judgments (see Taylor & Henson, 2012, for further discussion). While the latter effect may depend on precise experimental conditions (e.g. intermixed repetition and conceptual primes in same block), and certainly needs further replication, I think some of these papers should at least be cited before dismissing this Jacoby & Whitehouse paradigm as providing evidence in favour of dual-processes.

6. In the Introduction to Experiment 2, the rationale for the LoP manipulation is unclear. Encouraging deeper (semantic) processing of study items might improve their recognition memory in general, and memory for “internal” source (like the orienting task), but it is unclear why the authors ever thought it would improve memory for non-semantic, task-irrelevant information like temporal and spatial context (given that participants did not know that temporal and spatial source would later be tested)? Also, the rationale for using two orienting tasks for each level (deep and shallow) was not given (did the authors originally intend to test internal source memory too, ie memory for the specific orienting task?)

7. The rationale for other changes between Experiments 1 and 2 should also be stated in the Introduction to Experiment 2, eg differences in the response procedure and response options.

8. Some aspects of their results (eg in Experiment 2a and 2b) seem more consistent with a dual-process model (eg lack of an effect of LoP on priming), so might be given a bit more weight in the General Discussion.

9. I don’t understand why a high correlation between the two processes of the MS2 model indicates that “any influence of correlations across items at encoding (e.g., as a result of attentional fluctuation) is minimal” (page 44)?

10. What was justification for 44 participants in Exp 3b (e.g., in terms of power)? (It would seem powerful enough, but just wondering if there was a rationale)

I downloaded the data file and instructions provided by the authors (thanks). I did not check all the stats (this would take some time, and ideally the authors would provide the analysis scripts they used too), but I note that they only provide the mean identification times for each condition in each experiment; they do not include the number of trials in each condition (over which RTs

were averaged), nor ideally data for every single trial (eg in “long format”). The latter could be used for the above suggestions of (future) application of linear mixed effects models and (future) application of their previous computational model, which was fit to single trials (Berry et al, 2012).

A final postscript: in future, it is much easier to review if figures are inserted inline with the text, rather than gathered at the end. Most journals allow this, these days.

I declared to the editor that I previously worked with the senior author on the same basic topic, and she said this was ok as long as I could be unbiased (which I have tried to be) and was not actively collaborating (which I am not).

Refs

Woollams, A., Taylor, J.R., Karayanidis, F. & Henson, R.N. (2008). ERPs associated with masked priming of test cues reveal multiple potential contributions to recognition memory. *Journal of Cognitive Neuroscience*, 20, 1114-1129.

Taylor, J.R., Buratto, L. & Henson, R.N. (2013). Behavioural and neural evidence for masked conceptual priming of recollection. *Cortex*, 49, 1511-1525.

Taylor, J.R. & Henson, R.N. (2012). Could masked conceptual primes increase recollection? The subtleties of measuring recollection and familiarity in recognition memory. *Neuropsychologia*, 50, 3027-3040.

Review form: Reviewer 2

Is the manuscript scientifically sound in its present form?

Yes

Are the interpretations and conclusions justified by the results?

Yes

Is the language acceptable?

Yes

Do you have any ethical concerns with this paper?

No

Have you any concerns about statistical analyses in this paper?

No

Recommendation?

Major revision is needed (please make suggestions in comments)

Comments to the Author(s)

This is an interesting manuscript that presents work testing a core assumption of the dual process model, that fluency should be related to recognition memory but not to source memory. The authors present a number of experiments suggesting that's not the case – greater priming was found for items that were accompanied by accurate source decisions. These findings really undermine several assumptions of the dual process model along with theories that posit separations between implicit and explicit memory. This work strikes me as publishable. I have a number of issues that I'll document below, but I believe these can be addressed in a revision.

Point #1: Analysis

In terms of analyses, the authors have used null hypothesis significance testing (NHST) throughout the manuscript. A downside of this approach is that there are a few points throughout where non-significant results were obtained, and it's unclear whether this is because the null hypothesis is in fact true or whether there just wasn't enough data to decide between the null and alternative hypotheses. Bayesian ANOVAs, t-tests, and correlations can all be easily done nowadays with JASP. I strongly recommend that the authors consider substituting the present analyses with Bayesian analyses and interpreting the results accordingly.

Part of my concern here is that the 95% confidence intervals in several of the studies appear to be somewhat large (to my eyes at least!). Some of the experiments do not collect a lot of data per participant (Experiment 2A only collected 15 minutes of data per participant). It's possible that some of the non-significant differences found were really due to noise. What sticks out to me was the result in Experiment 1 where there was no difference found between the "both correct" and "one correct" identification latencies.

However, I also want to be explicit that while I find the analysis could be improved using Bayesian methods, the non-significant results found in the paper were largely non-central. So while I think Bayesian methods can improve the paper, I'm certainly not suggesting that the paper is currently "broken."

Hit and false alarm counts were analyzed throughout instead of hit and false alarm rates. Where I'm somewhat concerned about this is that subjects will have different numbers of target and lure trials because of the exclusions of misidentification trials. It seems like it would be better to conduct the analyses on rates instead for this reason, as this should reduce the variance across subjects. However, I really could be wrong here, so if there is good reason to analyze counts instead of rates that reasoning should be made explicit in the paper.

Point #2: A comparison should be performed to the Lange et al. (2019) paper

In terms of writing, there are some parts that are unclear and there is an incomplete literature review. I was really quite surprised to not see the following paper discussed:

Lange, N., Berry, C. J., & Hollins, T. J. (2019). Linking repetition priming, recognition, and source memory: A single-system signal-detection account. *Journal of Memory and Language*, 109.

Lange et al. conducted an investigation that was really quite similar to the present one, where identification latencies were compared between item recognition and source memory as a test of the dual process model. Furthermore, they also compared different signal detection models to data.

I want to be explicit that I don't believe the similarity between the Lange et al. work and the present manuscript merits rejection, as having experiments that replicate and extend the Lange et al. work are a valuable contribution to the literature. However, I do believe this merits some compare-and-contrast. What's different about the current set of experiments?

Point #3: Unclear methodological details

I wasn't clear on how the levels of processing manipulation was conducted in Experiment 2A. From the description, it sounds like the task varied from trial-to-trial, meaning that on one trial a deep processing task might be required and on the next a shallow processing task might be required.

However, in the analysis, false alarms were compared across the deep and shallow tasks. If the list was a mixed list, lures don't have any association to the tasks (unless they were similar to words from one of the tasks, was that the case?). Note that this concern doesn't apply to "pure" study lists where only one task was performed – in this case the false alarm rate is just associated with the entire study list.

Point #4: Varied experimental goals should be outlined in the introduction

This was somewhat of a meandering paper, because the introduction made it seem like the primary purpose of the experiments was solely to test the relationship between processing fluency and source memory. However, along the way a whole host of other secondary goals arose in subsequent experiments, such as levels of processing manipulations, temporal order memory, and fan manipulations. The introduction could do a better job of setting up these myriad sub-goals, perhaps with an additional paragraph or two outlining the manipulations employed in Experiments 2 and 3. Likewise, I think the General Discussion could come back to some of these subgoals as well.

Point #5: More complete data required in the repository

I've checked the data in the repository and it appears that only summary statistics are provided for each participant, namely mean RTs. I strongly recommend that the authors instead provide the complete dataset, namely all of the individual trials (ideally from the study phase as well). This allows researchers to perform analyses that are different from those performed by the original authors and in some cases to apply computational models to the data.

More minor comments:

There were a few other papers that merited some discussion in the introduction or general discussion. In particular, I think the Wixted and Mickes's (2011, Psychological Review) continuous dual process model merits some discussion along with the experiments they conducted to test it. In their experiments, they collected *both* confidence and remember/know (RK) judgments. They found that while "know" responses can be just as accurate as "remember" responses in item recognition if you equate confidence, the same was NOT true for source memory. Specifically, R judgments were much more uniquely predictive of source memory accuracy. This struck me as a compelling case that RK judgments are measuring something different than what the single-system view argues.

In addition, when discussing the RK judgments in the introduction, it strikes me as relevant to also mention the Dunn (2008, Psychological Review) state-trace analysis work. This presented a compelling case that R judgments are related to the old-new hit rate in a monotonic but non-linear fashion, which is what would be predicted by SDT. There was no evidence that two latent variables distinguished R from K judgments (in item recognition at least).

Finally, when discussing single-system views, it seems relevant to also cite some work using global matching models, which also would predict strong relationships between item recognition and source memory. Osth, Fox, McKague, Dennis, and Heathcote (2019, Journal of Memory and Language) developed source memory models from both the Osth and Dennis (2015) model and the REM model (Shiffrin & Steyvers, 1997) – in each model the only thing that distinguishes item recognition from source memory is that additional source features are used in the retrieval cues. I realize I'm asking for a citation to my own work here, but global matching models and their interpretations often get ignored in literature reviews.

The third paragraph kind of dives into some descriptions of the dual process model without actually defining what it is (this doesn't come later until the fourth paragraph). I suspect this won't make sense to some readers.

In the first paragraph: "In contrast to source memory, item memory refers to memory of a focal stimulus or event, but does not encompass its spatiotemporal context or associated features." I realize I'm being knipicky, but it's more fair to say that item memory doesn't *require* spatiotemporal context or features. Several models of recognition memory claim that items are encoded in context, including the cited Anderson and Bower (1972) model.

On p19, there is some description of how the format of R and K instructions could affect results and how "familiar" response options might change things. However, I didn't really understand how this was theoretically relevant to the present investigation. It's possible that I'm just not up on this literature, but it seems like this should either be cut or some more elaboration could be given.

The introduction to Experiment 2A claims that not a lot of studies have conducted depth of processing manipulations with source memory, but doesn't cite Glanzer, Hilford, and Kim (2004, *Journal of Experimental Psychology: Learning, Memory, and Cognition*). This paper conducts several manipulations on item recognition and source memory and their Experiment 3 found benefits of deep processing for both item recognition and source memory.

Decision letter (RSOS-190430.R0)

Dear Miss Huang

The Editors assigned to your paper RSOS-190430 "Examining the Relationship between Processing Fluency and Memory for Source Information" have now received comments from reviewers and would like you to revise the paper in accordance with the reviewer comments and any comments from the Editors. Please note this decision does not guarantee eventual acceptance.

Please submit your revised manuscript and required files (see below) no later than 21 days from today's (ie 27-Aug-2020) date. Note: the ScholarOne system will 'lock' if submission of the

revision is attempted 21 or more days after the deadline. If you do not think you will be able to meet this deadline please contact the editorial office immediately.

on behalf of Dr Alexa Morcom (Associate Editor) and Essi Viding (Subject Editor)
openscience@royalsociety.org

Associate Editor Comments to Author (Dr Alexa Morcom):

Associate Editor: 1

Comments to the Author:

Dear Ms Huang

Thank you for submitting your manuscript, "Examining the Relationship between Processing Fluency and Memory for Source Information", to Royal Society Open Science. We have now received two independent reviews of the work. Although the reviewers both found the results interesting, they have raised several substantive concerns which preclude publication in RSOS, at least in its present form.

While all the reviewers' comments merit careful consideration in a revision, three stand out. Reviewer 1 (Henson) raises concerns about the limitations of what can be learned from associations such as those reported here. Reviewer 2 (Osth) makes several important points about the writing of the manuscript, most notably its place in the literature given recent publication of a similar study by Lange et al. (2019) which is not cited. He also suggests that efforts be made, for example using Bayes Factors, to clarify which null findings are decisive and which are not. I also note that both reviewers mention that only summary data have so far been made available – please remedy this in your resubmission.

The reviews are appended to this message. We hope that you will be able to address the reviewers' concerns in full and resubmit the manuscript, along with a point-by-point reply to the reviews that indicates your response to each concern. Before we make a decision about publication, we will have your revision re-reviewed.

Yours sincerely
Alexa Morcom

Associate Editor: 2

Comments to the Author:
(There are no comments.)

Reviewer comments to Author:

Reviewer: 1

Comments to the Author(s)

This paper describes a series of behavioural experiments testing the relationship between fluency, as measured by identification times in a perceptual priming task, and memory for various types of source, using the CID-R paradigm previously employed by the authors' group. While the conclusions remain unclear, and it is a shame that the authors did not apply their published computational model, the experiments are well-conducted and powered, so will hopefully provide constraints on future theories and models.

1. The main finding is that correct source judgments are associated with faster identification times than incorrect judgments, and the (implicit) aim of the authors is to claim that this supports single-process models in which priming and source memory emerge from the same memory signal, contrary to the conventional view that implicit and explicit (certainly episodic) memory reflect different memory processes/systems. While I think it is good to question conventional wisdom, I am generally more convinced by dissociations than associations. There are numerous behavioural and neuroscientific dissociations between priming and episodic memory, and while I accept that many of these have faults, and that all single dissociations can be trivially explained by a single-process model, there are a few key findings (such as the reversed association of Richardson-Klavehn et al, 1999) that I think refute single-process models. By contrast, there are many trivial reasons for associations to emerge, even from multi-process models. For example, "random" fluctuations in attention from trial-to-trial during the study phase are likely to affect processes that underlie both fluency and recollection. In other words, correlations between two measures (priming and source memory) could reflect a single cause (attention) having similar effects on two, independent processes (fluency and recollection). This was on my mind throughout reading the paper, so it was gratifying to see the authors acknowledge this in the General Discussion. However, this point could have been made earlier, and the language could be modified to reflect this in several places; for example in the concluding paragraph, the authors say their study is "the first to reveal directly that familiarity-based processes influence source memory accuracy", but they have no direct evidence that familiarity *influences* (in causal sense) source memory, only that there are conditions under which the two are correlated (possibly because of a cause having parallel effects on two processes). Likewise for earlier statements like: "These results demonstrate that familiarity can contribute to R decisions, and contradict the interpretation of R responses as a process-pure index of recollection" - they do not demonstrate this directly.

2. In the absence of dissociations, a better way to compare single- vs dual-process theories is to fit formal models to the data, and ask whether the extra complexity of dual-processes models is justified by the data (i.e, an argument based on parsimony). The authors have done this elegantly in previous work (as they describe here in General Discussion) - but was there a reason why they didn't compare models in their ability (e.g, in terms of AIC/BIC) to fit the present data? Also, it's not clear to me why the authors focus here on their prior argument that the parameter that captures the correlation between the processes associated with priming and recognition memory was previously high (.93). I don't think this is as good evidence for a single-process model as formal measures of (penalized) model fit (e.g., AIC/BIC or cross-validation) and tests of model recovery/mimicry - if a dual-process model with high correlation between processes were still favoured by the data, then it would not matter if this correlation were high (perhaps some of the remaining $(1-.93^2)$ variance reflects important distinct variance in the data). More importantly, perhaps this parameter would be lower if fit to the present data, and perhaps it would be lower

still in paradigms other than the CID-R paradigm where the trial-by-trial combination of a perceptual identification test immediately followed by a source memory judgment is likely to maximise the influence of factors that have similar effects on multiple memory processes (e.g. random fluctuations in attention during test trials, as well as study trials).

3. I would not insist on this, but would it be worth fitting linear mixed effects models (eg using “lmer” in R) to single-trial data? Not only might this give higher power and the ability to model stimulus effects, but it might allow tests of whether individual trial identification times “predict” (using a logistic/binomial linking function) individual trial source accuracy, over and above other factors?

Minor points

4. Does the statement: “Since all participants have encountered countless words and images outside of the experimental session...” in the Introduction apply to meaningless stimuli used in many recognition memory studies to address this issue, like the fractals used by Voss & Paller? I appreciate that participants might still use pre-experimental knowledge to try to interpret such stimuli, so the distinction is not clear-cut, but this might be worth mentioning.

5. Do the criticisms of Rajaram’s study in terms of a “small sample size” and “potential type-2 error inflation” apply to the several replications, such as Woollams et al (2008) and Taylor et al (2012, 2013) (Refs below). Personally, I think the evidence is quite convincing that masked repetition primes immediately before test items increase K judgments more than R judgments. I accept that this is only a single dissociation, so not conclusive evidence for a single-process models. Furthermore, I accept the final criticism regarding exclusive versus parallel procedures for R/K judgments (of Higham & Vokey). Nonetheless, there have been reports of a double dissociation, whereby masked conceptual primes affect R but not K judgments (see Taylor & Henson, 2012, for further discussion). While the latter effect may depend on precise experimental conditions (e.g. intermixed repetition and conceptual primes in same block), and certainly needs further replication, I think some of these papers should at least be cited before dismissing this Jacoby & Whitehouse paradigm as providing evidence in favour of dual-processes.

6. In the Introduction to Experiment 2, the rationale for the LoP manipulation is unclear. Encouraging deeper (semantic) processing of study items might improve their recognition memory in general, and memory for “internal” source (like the orienting task), but it is unclear why the authors ever thought it would improve memory for non-semantic, task-irrelevant information like temporal and spatial context (given that participants did not know that temporal and spatial source would later be tested)? Also, the rationale for using two orienting tasks for each level (deep and shallow) was not given (did the authors originally intend to test internal source memory too, ie memory for the specific orienting task?)

7. The rationale for other changes between Experiments 1 and 2 should also be stated in the Introduction to Experiment 2, eg differences in the response procedure and response options.

8. Some aspects of their results (eg in Experiment 2a and 2b) seem more consistent with a dual-process model (eg lack of an effect of LoP on priming), so might be given a bit more weight in the General Discussion.

9. I don’t understand why a high correlation between the two processes of the MS2 model indicates that “any influence of correlations across items at encoding (e.g., as a result of attentional fluctuation) is minimal” (page 44)?

10. What was justification for 44 participants in Exp 3b (e.g., in terms of power)? (It would seem powerful enough, but just wondering if there was a rationale)

I downloaded the data file and instructions provided by the authors (thanks). I did not check all the stats (this would take some time, and ideally the authors would provide the analysis scripts they used too), but I note that they only provide the mean identification times for each condition in each experiment; they do not include the number of trials in each condition (over which RTs were averaged), nor ideally data for every single trial (eg in "long format"). The latter could be used for the above suggestions of (future) application of linear mixed effects models and (future) application of their previous computational model, which was fit to single trials (Berry et al, 2012).

A final postscript: in future, it is much easier to review if figures are inserted inline with the text, rather than gathered at the end. Most journals allow this, these days.

I declared to the editor that I previously worked with the senior author on the same basic topic, and she said this was ok as long as I could be unbiased (which I have tried to be) and was not actively collaborating (which I am not).

Refs

Woollams, A., Taylor, J.R., Karayanidis, F. & Henson, R.N. (2008). ERPs associated with masked priming of test cues reveal multiple potential contributions to recognition memory. *Journal of Cognitive Neuroscience*, 20, 1114-1129.

Taylor, J.R., Buratto, L. & Henson, R.N. (2013). Behavioural and neural evidence for masked conceptual priming of recollection. *Cortex*, 49, 1511-1525.

Taylor, J.R. & Henson, R.N. (2012). Could masked conceptual primes increase recollection? The subtleties of measuring recollection and familiarity in recognition memory. *Neuropsychologia*, 50, 3027-3040.

Reviewer: 2

Comments to the Author(s)

This is an interesting manuscript that presents work testing a core assumption of the dual process model, that fluency should be related to recognition memory but not to source memory. The authors present a number of experiments suggesting that's not the case - greater priming was found for items that were accompanied by accurate source decisions. These findings really undermine several assumptions of the dual process model along with theories that posit separations between implicit and explicit memory. This work strikes me as publishable. I have a number of issues that I'll document below, but I believe these can be addressed in a revision.

Point #1: Analysis

In terms of analyses, the authors have used null hypothesis significance testing (NHST) throughout the manuscript. A downside of this approach is that there are a few points throughout where non-significant results were obtained, and it's unclear whether this is because the null hypothesis is in fact true or whether there just wasn't enough data to decide between the null and alternative hypotheses. Bayesian ANOVAs, t-tests, and correlations can all be easily done nowadays with JASP. I strongly recommend that the authors consider substituting the present analyses with Bayesian analyses and interpreting the results accordingly.

Part of my concern here is that the 95% confidence intervals in several of the studies appear to be somewhat large (to my eyes at least!). Some of the experiments do not collect a lot of data per participant (Experiment 2A only collected 15 minutes of data per participant). It's possible that some of the non-significant differences found were really due to noise. What sticks out to me was

the result in Experiment 1 where there was no difference found between the “both correct” and “one correct” identification latencies.

However, I also want to be explicit that while I find the analysis could be improved using Bayesian methods, the non-significant results found in the paper were largely non-central. So while I think Bayesian methods can improve the paper, I’m certainly not suggesting that the paper is currently “broken.”

Hit and false alarm counts were analyzed throughout instead of hit and false alarm rates. Where I’m somewhat concerned about this is that subjects will have different numbers of target and lure trials because of the exclusions of misidentification trials. It seems like it would be better to conduct the analyses on rates instead for this reason, as this should reduce the variance across subjects. However, I really could be wrong here, so if there is good reason to analyze counts instead of rates that reasoning should be made explicit in the paper.

Point #2: A comparison should be performed to the Lange et al. (2019) paper

In terms of writing, there are some parts that are unclear and there is an incomplete literature review. I was really quite surprised to not see the following paper discussed:

Lange, N., Berry, C. J., & Hollins, T. J. (2019). Linking repetition priming, recognition, and source memory: A single-system signal-detection account. *Journal of Memory and Language*, 109.

Lange et al. conducted an investigation that was really quite similar to the present one, where identification latencies were compared between item recognition and source memory as a test of the dual process model. Furthermore, they also compared different signal detection models to data.

I want to be explicit that I don’t believe the similarity between the Lange et al. work and the present manuscript merits rejection, as having experiments that replicate and extend the Lange et al. work are a valuable contribution to the literature. However, I do believe this merits some compare-and-contrast. What’s different about the current set of experiments?

Point #3: Unclear methodological details

I wasn’t clear on how the levels of processing manipulation was conducted in Experiment 2A. From the description, it sounds like the task varied from trial-to-trial, meaning that on one trial a deep processing task might be required and on the next a shallow processing task might be required.

However, in the analysis, false alarms were compared across the deep and shallow tasks. If the list was a mixed list, lures don’t have any association to the tasks (unless they were similar to words from one of the tasks, was that the case?). Note that this concern doesn’t apply to “pure” study lists where only one task was performed – in this case the false alarm rate is just associated with the entire study list.

Point #4: Varied experimental goals should be outlined in the introduction

This was somewhat of a meandering paper, because the introduction made it seem like the primary purpose of the experiments was solely to test the relationship between processing fluency and source memory. However, along the way a whole host of other secondary goals arose in subsequent experiments, such as levels of processing manipulations, temporal order memory, and fan manipulations. The introduction could do a better job of setting up these

myriad sub-goals, perhaps with an additional paragraph or two outlining the manipulations employed in Experiments 2 and 3. Likewise, I think the General Discussion could come back to some of these subgoals as well.

Point #5: More complete data required in the repository

I've checked the data in the repository and it appears that only summary statistics are provided for each participant, namely mean RTs. I strongly recommend that the authors instead provide the complete dataset, namely all of the individual trials (ideally from the study phase as well). This allows researchers to perform analyses that are different from those performed by the original authors and in some cases to apply computational models to the data.

More minor comments:

There were a few other papers that merited some discussion in the introduction or general discussion. In particular, I think the Wixted and Mickes's (2011, *Psychological Review*) continuous dual process model merits some discussion along with the experiments they conducted to test it. In their experiments, they collected *both* confidence and remember/know (RK) judgments. They found that while "know" responses can be just as accurate as "remember" responses in item recognition if you equate confidence, the same was NOT true for source memory. Specifically, R judgments were much more uniquely predictive of source memory accuracy. This struck me as a compelling case that RK judgments are measuring something different than what the single-system view argues.

In addition, when discussing the RK judgments in the introduction, it strikes me as relevant to also mention the Dunn (2008, *Psychological Review*) state-trace analysis work. This presented a compelling case that R judgments are related to the old-new hit rate in a monotonic but non-linear fashion, which is what would be predicted by SDT. There was no evidence that two latent variables distinguished R from K judgments (in item recognition at least).

Finally, when discussing single-system views, it seems relevant to also cite some work using global matching models, which also would predict strong relationships between item recognition and source memory. Osth, Fox, McKague, Dennis, and Heathcote (2019, *Journal of Memory and Language*) developed source memory models from both the Osth and Dennis (2015) model and the REM model (Shiffrin & Steyvers, 1997) – in each model the only thing that distinguishes item recognition from source memory is that additional source features are used in the retrieval cues. I realize I'm asking for a citation to my own work here, but global matching models and their interpretations often get ignored in literature reviews.

The third paragraph kind of dives into some descriptions of the dual process model without actually defining what it is (this doesn't come later until the fourth paragraph). I suspect this won't make sense to some readers.

In the first paragraph: "In contrast to source memory, item memory refers to memory of a focal stimulus or event, but does not encompass its spatiotemporal context or associated features." I realize I'm being nitpicky, but it's more fair to say that item memory doesn't *require* spatiotemporal context or features. Several models of recognition memory claim that items are encoded in context, including the cited Anderson and Bower (1972) model.

On p19, there is some description of how the format of R and K instructions could affect results and how "familiar" response options might change things. However, I didn't really understand how this was theoretically relevant to the present investigation. It's possible that I'm just not up

on this literature, but it seems like this should either be cut or some more elaboration could be given.

The introduction to Experiment 2A claims that not a lot of studies have conducted depth of processing manipulations with source memory, but doesn't cite Glanzer, Hilford, and Kim (2004, *Journal of Experimental Psychology: Learning, Memory, and Cognition*). This paper conducts several manipulations on item recognition and source memory and their Experiment 3 found benefits of deep processing for both item recognition and source memory.

===PREPARING YOUR MANUSCRIPT===

===PREPARING YOUR REVISION IN SCHOLARONE===

Author's Response to Decision Letter for (RSOS-190430.R0)

See Appendix A.

RSOS-190430.R1 (Revision)

Review form: Reviewer 1

Is the manuscript scientifically sound in its present form?

No

Are the interpretations and conclusions justified by the results?

Yes

Is the language acceptable?

Yes

Do you have any ethical concerns with this paper?

No

Have you any concerns about statistical analyses in this paper?

Yes

Recommendation?

Accept with minor revision (please list in comments)

Comments to the Author(s)

I think the paper is much improved, particularly the Discussion. However, I have some residual straightforward, but important, further requests.

1. I downloaded the data from Dryad, but was disappointed not to see any statistical analysis code, eg in R – please provide that for a further review. Also, please do not share data files that require proprietary software like Excel – just save as a CSV file. Finally, there were several strange values in the Excel file like “#VALUE!” and “#N/A”. While “N/A” is fine to indicate missing data, please clarify the meaning of values prepended with a “#”.
2. The full paper PDF was not available until I asked the editor to re-upload. Once obtained, the PDF contained some formatting errors – eg on page 5, I see the phrase “Most dual-process accounts assume that familiarity is driven by a continuum of memory” repeated 7 times. I don’t know whether I should ignore all these repetitions, or where there is a missing paragraph; another reason why I think a second, quick review is necessary.
3. It is interesting that there are many cases when the frequentist p-value and Bayes Factor diverge. I have experience of p-values just below .05 when BF10 is little more than 1, but I find it surprising when, e.g. on page 17, p is quite small (.0075) but BF10<1 (0.51). It might be worth double-checking the R code and priors used by BayesFactor package?
4. Page 42: I think “Following the a priori power and minimal sample size calculation from Experiment 3b...” should read “Following the a priori power and minimal sample size calculation from Experiment 3a...”
5. Page 56: were random slopes allowed for source dimension, LoP and Experiment? Was the optimal random effects structure estimated first?
6. Page 51: I think the discussion of the Lange et al study really helps, particularly in allaying the concern that the high correlation between priming and source memory is an artefact of sequential decisions on the same cue in the standard CID-R paradigm. While separating the two types of decision (into separate test phases) is important to rule out shared effects (eg of attention) during retrieval (a la Jacoby & Whitehouse), I’m not sure how “This suggests that any influence of correlations across items at encoding (e.g., as a result of attentional fluctuation) is minimal” – couldn’t fluctuations at encoding still produce correlation between tests on the same item, even if those tests occur in different sessions?

7. While the Dryad hyperlink at the start of the PDF is correct, I think the link quoted in the final "Data Availability" section is incorrect.

Review form: Reviewer 2

Is the manuscript scientifically sound in its present form?

Yes

Are the interpretations and conclusions justified by the results?

Yes

Is the language acceptable?

Yes

Do you have any ethical concerns with this paper?

No

Have you any concerns about statistical analyses in this paper?

Yes

Recommendation?

Major revision is needed (please make suggestions in comments)

Comments to the Author(s)

Many of my suggestions for the paper have been addressed. However, unfortunately the major issue that prevents me from endorsing publication is that the Bayesian analyses included in the paper were not properly interpreted anywhere. It appears that the authors merely inserted the Bayes Factor (BF) results. However, there are a number of places throughout the manuscript where the BF results are in direct contradiction to the frequentist results and this is not brought up or discussed. For instance:

On p17: "There was a main effect of subjective old/new judgements on identification RTs... $p = .0075$, ..., $BF_{10} = .51$ "

Given that the $BF < 1$, this implies that the null hypothesis is more likely than the alternative hypothesis (although the evidence would be considered weak).

P 17: "Post-hoc t-tests with Bonferroni correction revealed that identification RTs were faster for trials with R than K responses, ..., $p = .0294$, ..., $BF = 1.03$, and K trials were associated with faster identification RTs compared to G trials, ..., $p = .0281$, $BF_{10} = 1.05$ "

In both of these t-tests, the $BF \sim 1$ implies that both the null and alternative hypotheses are equally likely, meaning that the data are completely ambiguous.

I won't elaborate on all of the cases within the manuscript, but it strikes me that the Results need to be re-interpreted and appropriately discussed in light of the Bayesian analyses. I also recommend some (brief) discussion of how BF results should be interpreted. Usually these are done with recourse to the standards of Jeffreys (1961), where any BFs between 1/3 and 3 are considered only "anecdotal" evidence for or against a particular hypothesis. I recommend consulting the work of EJ Wagenmakers or Jeff Rouder. This paper was particularly influential:

Wagenmakers, E. J. (2007). A practical solution to the pervasive problem of p-values. *Psychonomic Bulletin & Review*, 14, 779-804.

There are also a lot of great papers in the special issue of *Psychonomic Bulletin & Review* on Bayesian analyses.

Minor comments:

There is some discussion about reversed association results within. It may help to just **briefly** cite and mention the following paper which did not replicate some of these important results:

Haaf, J. M., Rhodes, S., Naveh-Benjamin, M. Sun, T., Snyder, K., & Rouder, J. N. (2020). Revisiting the remember-know task: Replications of Gardiner and Java (1990). *Memory & Cognition*.

The literature review is already quite long, so just a brief mention would suffice! Even a "but see..."

Decision letter (RSOS-190430.R1)

Dear Miss Huang

The Editors assigned to your paper RSOS-190430.R1 "Examining the Relationship between Processing Fluency and Memory for Source Information" have now received comments from reviewers and would like you to revise the paper in accordance with the reviewer comments and any comments from the Editors. Please note this decision does not guarantee eventual acceptance.

Please submit your revised manuscript and required files (see below) no later than 21 days from today's (ie 25-Jan-2021) date. Note: the ScholarOne system will 'lock' if submission of the revision is attempted 21 or more days after the deadline. If you do not think you will be able to meet this deadline please contact the editorial office immediately.

Please note article processing charges apply to papers accepted for publication in Royal Society Open Science (<https://royalsocietypublishing.org/rsos/charges>). Charges will also apply to

papers transferred to the journal from other Royal Society Publishing journals, as well as papers submitted as part of our collaboration with the Royal Society of Chemistry (<https://royalsocietypublishing.org/rsos/chemistry>). Fee waivers are available but must be requested when you submit your revision (<https://royalsocietypublishing.org/rsos/waivers>).

Best regards,

on behalf of Dr Alexa Morcom (Associate Editor) and Essi Viding (Subject Editor)
openscience@royalsociety.org

Associate Editor Comments to Author (Dr Alexa Morcom):

Dear Ms Huang

Thank you for resubmitting your revised manuscript, "Examining the Relationship between Processing Fluency and Memory for Source Information", to Royal Society Open Science. We have now received two further reviews from the original two independent reviewers. Although both find the manuscript to be substantially improved, significant concerns remain that preclude publication in RSOS in its present form.

The main point, raised by both reviewers, is that the frequentist p-values and Bayes Factor results appear to diverge in many places. This raises two issues: first, as Reviewer 1 (Henson) notes, there may be errors in the R code or priors used by the Bayes Factor package, and these should be checked. He was unable to examine the code himself and it is therefore important that this code also be made available as part of any resubmission. Second, as Reviewer 2 (Osth) notes, the results need to be appropriately interpreted in light of the Bayesian results, particularly where p-values are significant but Bayesian evidence is inconclusive. Henson also makes a few other important points; both reviews are appended to this message.

We hope that you will be able to address these further concerns in full and will be able to resubmit the manuscript. When you do, please include a point-by-point reply to each of the reviewers' concerns. Before we make a decision about publication, we will have your revision re-reviewed.

Reviewer comments to Author:

Reviewer: 1
Comments to the Author(s)

I think the paper is much improved, particularly the Discussion. However, I have some residual straightforward, but important, further requests.

1. I downloaded the data from Dryad, but was disappointed not to see any statistical analysis code, eg in R - please provide that for a further review. Also, please do not share data files that

require proprietary software like Excel – just save as a CSV file. Finally, there were several strange values in the Excel file like “#VALUE!” and “#N/A”. While “N/A” is fine to indicate missing data, please clarify the meaning of values prepended with a “#”.

2. The full paper PDF was not available until I asked the editor to re-upload. Once obtained, the PDF contained some formatting errors – eg on page 5, I see the phrase “Most dual-process accounts assume that familiarity is driven by a continuum of memory” repeated 7 times. I don’t know whether I should ignore all these repetitions, or where there is a missing paragraph; another reason why I think a second, quick review is necessary.

3. It is interesting that there are many cases when the frequentist p-value and Bayes Factor diverge. I have experience of p-values just below .05 when BF10 is little more than 1, but I find it surprising when, e.g. on page 17, p is quite small (.0075) but BF10 < 1 (0.51). It might be worth double-checking the R code and priors used by BayesFactor package?

4. Page 42: I think “Following the a priori power and minimal sample size calculation from Experiment 3b...” should read “Following the a priori power and minimal sample size calculation from Experiment 3a...”

5. Page 56: were random slopes allowed for source dimension, LoP and Experiment? Was the optimal random effects structure estimated first?

6. Page 51: I think the discussion of the Lange et al study really helps, particularly in allaying the concern that the high correlation between priming and source memory is an artefact of sequential decisions on the same cue in the standard CID-R paradigm. While separating the two types of decision (into separate test phases) is important to rule out shared effects (eg of attention) during retrieval (a la Jacoby & Whitehouse), I’m not sure how “This suggests that any influence of correlations across items at encoding (e.g., as a result of attentional fluctuation) is minimal” – couldn’t fluctuations at encoding still produce correlation between tests on the same item, even if those tests occur in different sessions?

7. While the Dryad hyperlink at the start of the PDF is correct, I think the link quoted in the final “Data Availability” section is incorrect.

Reviewer: 2

Comments to the Author(s)

Many of my suggestions for the paper have been addressed. However, unfortunately the major issue that prevents me from endorsing publication is that the Bayesian analyses included in the paper were not properly interpreted anywhere. It appears that the authors merely inserted the Bayes Factor (BF) results. However, there are a number of places throughout the manuscript where the BF results are in direct contradiction to the frequentist results and this is not brought up or discussed. For instance:

On p17: "There was a main effect of subjective old/new judgements on identification RTs... p = .0075, ..., BF_10 = .51"

Given that the BF < 1, this implies that the null hypothesis is more likely than the alternative hypothesis (although the evidence would be considered weak).

P 17: "Post-hoc t-tests with Bonferroni correction revealed that identification RTs were faster for trials with R than K responses, ..., p = .0294, ..., BF = 1.03, and K trials were associated with faster identification RTs compared to G trials, ..., p = .0281, BF_10 = 1.05"

In both of these t-tests, the BF ~ 1 implies that both the null and alternative hypotheses are equally likely, meaning that the data are completely ambiguous.

I won't elaborate on all of the cases within the manuscript, but it strikes me that the Results need to be re-interpreted and appropriately discussed in light of the Bayesian analyses. I also recommend some (brief) discussion of how BF results should be interpreted. Usually these are done with recourse to the standards of Jeffreys (1961), where any BFs between 1/3 and 3 are considered only "anecdotal" evidence for or against a particular hypothesis. I recommend consulting the work of EJ Wagenmakers or Jeff Rouder. This paper was particularly influential:

Wagenmakers, E. J. (2007). A practical solution to the pervasive problem of p-values. *Psychonomic Bulletin & Review*, 14, 779-804.

There are also a lot of great papers in the special issue of *Psychonomic Bulletin & Review* on Bayesian analyses.

Minor comments:

There is some discussion about reversed association results within. It may help to just **briefly** cite and mention the following paper which did not replicate some of these important results:

Haaf, J. M., Rhodes, S., Naveh-Benjamin, M. Sun, T., Snyder, K., & Rouder, J. N. (2020). Revisiting the remember-know task: Replications of Gardiner and Java (1990). *Memory & Cognition*.

The literature review is already quite long, so just a brief mention would suffice! Even a "but see..."

===PREPARING YOUR MANUSCRIPT===

If you have been asked to revise the written English in your submission as a condition of publication, you must do so, and you are expected to provide evidence that you have received language editing support. The journal would prefer that you use a professional language editing service and provide a certificate of editing, but a signed letter from a colleague who is a native

speaker of English is acceptable. Note the journal has arranged a number of discounts for authors using professional language editing services (<https://royalsociety.org/journals/authors/benefits/language-editing/>).

===PREPARING YOUR REVISION IN SCHOLARONE===

<https://royalsociety.org/journals/authors/author-guidelines/#supplementary-material> to include a suitable title and informative caption. An example of appropriate titling and captioning may be found at https://figshare.com/articles/Table_S2_from_Is_there_a_trade-

off_between_peak_performance_and_performance_breadth_across_temperatures_for_aerobic_sc
ope_in_teleost_fishes_/3843624.

Author's Response to Decision Letter for (RSOS-190430.R1)

See Appendix B.

Decision letter (RSOS-190430.R2)

Dear Miss Huang,

It is a pleasure to accept your manuscript entitled "Examining the Relationship between Processing Fluency and Memory for Source Information" in its current form for publication in Royal Society Open Science. The comments from the Editors are included at the foot of this letter.

on behalf of Dr Alexa Morcom (Associate Editor) and Essi Viding (Subject Editor)

Associate Editor Comments to Author (Dr Alexa Morcom):

Dear Ms Huang

Thank you for resubmitting this further revision of your manuscript, "Examining the Relationship between Processing Fluency and Memory for Source Information", to Royal Society Open Science.

I appreciate your responsiveness to the reviewers' suggestions and am delighted to inform you that we can now accept the paper in its current form.

I will now pass your paper to our editorial team who will take it to the production stage. I look forward to seeing it published and to receiving further submissions from you in the future.

Appendix A

RSOS-190430

Reviewer #1			
Major points	Reviewer comment	Author response/Actions	Page reference
1.1	...while I accept that many of these have faults, and that all single dissociations can be trivially explained by a single-process model, there are a few key findings (such as the reversed association of Richardson-Klavehn et al, 1999) that I think refute single-process models.	We acknowledge the limitation of the single-system model in explaining reversed association findings, and have added Gardiner (2000) and Richardson-Klavehn et al (1999) as examples.	6
1.2	...in other words, correlations between two measures (priming and source memory) could reflect a single cause (attention) having similar effects on two, independent processes (fluency and recollection) ... this point could have been made earlier, and the language could be modified to reflect this in several places; for example in the concluding paragraph, the authors say their study is “the first to reveal directly that familiarity-based processes influence source memory accuracy”, but they have no direct evidence that familiarity *influences* (in causal sense) source memory, only that there are conditions under which the two are correlated (possibly because of a cause having parallel effects on two processes).” Likewise for earlier statements like: “These results demonstrate that familiarity can contribute to R decisions, and contradict the interpretation of R responses as a process-pure index of recollection” – they do not demonstrate this directly.	We now note earlier that the possibility that the priming-source memory association may have originated from a single cause which produces similar effects on fluency and recollection as separate processes. We modify the language of statements which appeared to imply a direct causal influence of familiarity on source memory.	21-22 2, 20, 22, 48, 53
2.1	In the absence of dissociations, a better way to compare single- vs dual-process theories is to fit formal models to the data, and ask whether the extra complexity of dual-processes models is justified by the data (i.e, an argument based on parsimony) ... was there a reason why [the authors] didn't compare models in their ability (e.g, in terms of AIC/BIC) to fit the present data?	We agree that it would be highly valuable to compare the single vs. dual/multiple systems models in their ability to fit the data. Simply because of the scale of work involved, it is beyond the scope and design of the present study, whose primary purpose was an empirical exploration (though with model-fitting to follow). We have added a further discussion of the implications for/of a computational	49-52

		investigative approach, as well as recent work by Lange et al. (2019) which showed that a single-system model provides the most parsimonious overall fit to their CID-R source memory data.	
2.2	it's not clear to me why the authors focus here on their prior argument that the parameter that captures the correlation between the processes associated with priming and recognition memory was previously high (.93). I don't think this is as good evidence for a single-process model as formal measures of (penalized) model fit (e.g., AIC/BIC or cross-validation) and tests of model recovery/mimicry – if a dual-process model with high correlation between processes were still favoured by the data, then it would not matter if this correlation were high (perhaps some of the remaining $(1-.93^2)$ variance reflects important distinct variance in the data) ... perhaps this parameter would be lower if fit to the present data, and perhaps it would be lower still in paradigms other than the CID-R paradigm where the trial-by-trial combination of a perceptual identification test immediately followed by a source memory judgment is likely to maximise the influence of factors that have similar effects on multiple memory processes (e.g, random fluctuations in attention during test trials, as well as study trials)	We acknowledge that, without having applied formal models to our data, we cannot prove that the correlation parameter between priming and source/recognition memory is as high as .93. We refer to recent and independent findings from Lange et al. (2019) which suggested that, in terms of ability to fit their source memory data derived from a similar CID-R paradigm, a multiple-systems model with a low correlation parameter (MS1) is inferior to both the MS2 model with a high correlation parameter and the single-system model. We additionally mention that, although the association between priming and recognition memory weakened in their Expts 3a and 3b (which split the identification test and memory judgements into two separate phases) the associations between identification RTs, source memory accuracy, and source confidence still persisted.	50-51
3	... would it be worth fitting linear mixed effects models (eg using “lmer” in R) to single-trial data? Not only might this give higher power and the ability to model stimulus effects, but it might allow tests of whether individual trial identification times “predict” (using a logistic/binomial linking function) individual trial source accuracy, over and above other factors?	We have taken up this excellent suggestion, for which we thank the reviewer, and added mixed-effects analysis results. The results strongly confirm our conclusions, showing that the association between source accuracy and identification RTs may persist at the individual trial level.	11, 46-47, Tables 2&3 (75-77)
Minor points			

4	Does the statement: “Since all participants have encountered countless words and images outside of the experimental session...” in the Introduction apply to meaningless stimuli used in many recognition memory studies to address this issue, like the fractals used by Voss & Paller? I appreciate that participants might still use pre-experimental knowledge to try to interpret such stimuli, so the distinction is not clear-cut, but this might be worth mentioning.	Although recognition memory tests commonly employ ‘meaningful’ stimuli (e.g., word lists), we agree that the distinction between meaningless and meaningful stimuli in inter-/extra-experimental settings is not clear cut, especially in the case of novel stimuli such as the fractal example. We have now added this point to the statement.	4
5.1	Do the criticisms of Rajaram’s study in terms of a “small sample size” and “potential type-2 error inflation” apply to the several replications, such as Woollams et al (2008) and Taylor et al (2012, 2013) (Refs below). Personally, I think the evidence is quite convincing that masked repetition primes immediately before test items increase K judgments more than R judgments. I accept that this is only a single dissociation, so not conclusive evidence for a single-process models.	We agree that Woollams et al (2008 and Taylor et al (2012,2013) have replicated the finding that priming increases K judgements to a greater degree than R judgements. As those studies were well powered, the criticisms of Rajaram’s study did not apply to them. We have now added this point following the discussion on Rajaram.	8
5.2	I accept the final criticism regarding exclusive versus parallel procedures for R/K judgments (of Higham & Vokey). Nonetheless, there have been reports of a double dissociation, whereby masked conceptual primes affect R but not K judgments (see Taylor & Henson, 2012, for further discussion). While the latter effect may depend on precise experimental conditions (e.g, intermixed repetition and conceptual primes in same block), and certainly needs further replication, I think some of these papers should at least be cited before dismissing this Jacoby & Whitehouse paradigm as providing evidence in favour of dual-processes.	We now cite these double dissociations, such as in Taylor et al (2012, 2013), and acknowledge that they provide stronger evidence in favour of dual-process models, as compared to single dissociation findings e.g., Jacoby & Whitehouse or Rajaram.	8-9
6.1	In the Introduction to Experiment 2, the rationale for the LoP manipulation is unclear. Encouraging deeper (semantic) processing of study items might improve their	We elaborate more on the LoP manipulation and state the rationale for the manipulation (final paragraph of the Expt 2a introduction), which was to	23-25

	recognition memory in general, and memory for “internal” source (like the orienting task), but it is unclear why the authors ever thought it would improve memory for non-semantic, task irrelevant information like temporal and spatial context (given that participants did not know that temporal and spatial source would later be tested)?	explore whether LoP enhancements to memory also extend to external/task-irrelevant contextual information.	
6.2	...the rationale for using two orienting tasks for each level (deep and shallow) was not given (did the authors originally intend to test internal source memory too, ie memory for the specific orienting task?)	We never intended to test internal source memory for the specific orienting questions. Two questions orienting questions were used per level to ensure construct validity of the task. Clarification and rationale are now added in the introduction to Expt 2a.	26
7	The rationale for other changes between Experiments 1 and 2 should also be stated in the Introduction to Experiment 2, eg differences in the response procedure and response options.	Rationales for these alterations are now included.	27
8	Some aspects of their results (eg in Experiment 2a and 2b) seem more consistent with a dual-process model (eg lack of an effect of LoP on priming), so might be given a bit more weight in the General Discussion.	We now mention the absence of LoP effects on priming in Expts 2a and 2b, and how this may be more consistent with a dual-process model as opposed to a single process model.	51-52
9	I don't understand why a high correlation between the two processes of the MS2 model indicates that “any influence of correlations across items at encoding (e.g., as a result of attentional fluctuation) is minimal” (page 44)?	This sentence was an error and has now been removed. A paragraph beginning later on page 48 discusses the issue of correlations across items possibly arising from attentional fluctuations during encoding.	51
10	What was justification for 44 participants in Exp 3b (e.g., in terms of power)? (It would seem powerful enough, but just wondering if there was a rationale)	This was based on the sample size and a priori power calculation of Expt 3a. Explanation is now added to the Participants subsection of Expt 3b.	42
Misc.	Availability of trial-by-trial data	We re-uploaded a version of the data in long-format on the Dryad open repository, with single trial data for all the experiments.	

Reviewer #2			
Major points	Reviewer comment	Author response/Actions	Page reference
1.1	... the authors have used null hypothesis significance testing (NHST) throughout the manuscript. A downside of this approach is that there are a few points throughout where non-significant results were obtained, and it's unclear whether this is because the null hypothesis is in fact true or whether there just wasn't enough data to decide between the null and alternative hypotheses. Bayesian ANOVAs, t-tests, and correlations can all be easily done nowadays with JASP. I strongly recommend that the authors consider substituting the present analyses with Bayesian analyses and interpreting the results accordingly	We thank the reviewer for suggesting the use of Bayes factors, and we have now reported them along with our main analyses, and wherever else possible. However, we decided to also keep the values from the frequentist analyses in the manuscript, as they were part of, and had informed, the original designs of our experiments (e.g., a priori sample size planning).	17-19, 29-31, 35-36, 40-41, 44
1.2	Part of my concern here is that the 95% confidence intervals in several of the studies appear to be somewhat large (to my eyes at least!). Some of the experiments do not collect a lot of data per participant (Experiment 2A only collected 15 minutes of data per participant). It's possible that some of the non-significant differences found were really due to noise.	We acknowledge that some experiments, especially 2a, did not collect a lot of data per participant, and that this could have contributed to noise in the data. Expt 2A was especially short per participant, due to the constraints of being conducted in a classroom setting. In terms of practicality, it was also difficult to extend the testing of the other experiments beyond 40-50 minutes, as we avoided causing participant fatigue due to the repetitive nature of the task.	27, 33
1.3	What sticks out to me was the result in Experiment 1 where there was no difference found between the "both correct" and "one correct" identification latencies.	We acknowledge this observation, and have also mentioned this in the discussion for Expt 1.	22-23
1.4	Hit and false alarm counts were analyzed throughout instead of hit and false alarm rates. Where I'm somewhat concerned about this is that subjects will have different numbers of target and lure trials because of the exclusions of misidentification trials. It seems like it would be better to conduct the analyses on rates instead for this	We have now changed hit and false alarm rate analyses from counts to rates. We had reported hit and false alarm rates for all experiments. Except in Expt 2a, no analyses were carried out on (item level) hit and false alarm counts, nor rates, as those were not relevant to the source memory analyses.	29-30

	reason, as this should reduce the variance across subjects.		
2.1	In terms of writing, there are some parts that are unclear and there is an incomplete literature review. I was really quite surprised to not see the [Lange et al. (2019)] paper discussed	Following the reviewers' helpful recommendations, we clarified the text and discussed additional relevant literature, including the Lange et al. (2019).	50-51
2.2	... I don't believe the similarity between the Lange et al. work and the present manuscript merits rejection, as having experiments that replicate and extend the Lange et al. work are a valuable contribution to the literature. However, I do believe this merits some compare-and-contrast. What's different about the current set of experiments?	The experiments in the present manuscripts were conducted at about the same time as Lange et al.'s, and more importantly (as we have noted in the manuscript) conducted independently of their group. Our experiments focused on other factors (e.g., LoP, modality, and contextual fan) which may moderate source memory, which adds to Lange et al.'s investigation of how list length (Expt 2) moderates the association between priming and source memory.	50-52
3.1	I wasn't clear on how the levels of processing manipulation was conducted in Experiment 2A. From the description, it sounds like the task varied from trial-to-trial, meaning that on one trial a deep processing task might be required and on the next a shallow processing task might be required.	Yes, the processing task varied from trial-to-trial during encoding. We now clarify this detail under the Design and Materials subsection of Experiment 2a.	26
3.2	... in the analysis, false alarms were compared across the deep and shallow tasks. If the list was a mixed list, lures don't have any association to the tasks (unless they were similar to words from one of the tasks, was that the case?). Note that this concern doesn't apply to "pure" study lists where only one task was performed – in this case the false alarm rate is just associated with the entire study list.	Yes, the lures were qualitatively similar to the words from the tasks, and they were similarly presented under each of those tasks during encoding. This is also now clarified under Design and Materials, Experiment 2a.	27
4	This was somewhat of a meandering paper, because the introduction made it seem like the primary purpose of the experiments was solely to test the relationship between processing fluency and source memory. However, along the way a whole host of other secondary	We mention in the general introduction the secondary interests of the experiments. In the General Discussion, we have now summarized and briefly discussed these secondary findings.	11, 49

	goals arose in subsequent experiments, such as levels of processing manipulations, temporal order memory, and fan manipulations. The introduction could do a better job of setting up these myriad sub-goals, perhaps with an additional paragraph or two outlining the manipulations employed in Experiments 2 and 3. Likewise, I think the General Discussion could come back to some of these subgoals as well.		
5	I've checked the data in the repository and it appears that only summary statistics are provided for each participant, namely mean RTs. I strongly recommend that the authors instead provide the complete dataset, namely all of the individual trials (ideally from the study phase as well). This allows researchers to perform analyses that are different from those performed by the original authors and in some cases to apply computational models to the data.	We re-uploaded a version of the data in long-format on the Dryad open repository, with single trial data for all the experiments.	
Minor Points			
6	There were a few other papers that merited some discussion in the introduction or general discussion. In particular, I think the Wixted and Mickes's (2011, Psychological Review) continuous dual process model merits some discussion along with the experiments they conducted to test it...	We included a discussion of Wixted and Mickes' continuous dual-process model, as well as experiments related to this (Tuney et al. 2012, Frontiers in Psychology).	7-8
7	... when discussing the RK judgments in the introduction, it strikes me as relevant to also mention the Dunn (2008, Psychological Review) state-trace analysis work. This presented a compelling case that R judgments are related to the old-new hit rate in a monotonic but non-linear fashion, which is what would be predicted by SDT. There was no evidence that two latent variables distinguished R from K judgments (in item recognition at least).	We incorporated a description of the findings of Dunn (2008) into the discussion on single-system interpretations of R/K judgements.	6

8	when discussing single-system views, it seems relevant to also cite some work using global matching models, which also would predict strong relationships between item recognition and source memory. Osth, Fox, McKague, Dennis, and Heathcote (2019, Journal of Memory and Language) developed source memory models from both the Osth and Dennis (2015) model and the REM model (Shiffrin & Steyvers, 1997) – in each model the only thing that distinguishes item recognition from source memory is that additional source features are used in the retrieval cues.	We added a discussion of the Osth et al. global matching model of source memory, as well as the model implications of their list strength effect findings.	6-7
9	The third paragraph kind of dives into some descriptions of the dual process model without actually defining what it is (this doesn't come later until the fourth paragraph). I suspect this won't make sense to some readers.	We merged paragraph 4 into paragraph 3.	4
10	In the first paragraph: "In contrast to source memory, item memory refers to memory or a focal stimulus or event, but does not encompass its spatiotemporal context or associated features." ... it's more fair to say that item memory doesn't *require* spatiotemporal context or features. Several models of recognition memory claim that items are encoded in context, including the cited Anderson and Bower (1972) model.	We agree that there should be more nuance in this statement, and we changed "does not encompass" to "does not require".	3
11	On p19, there is some description of how the format of R and K instructions could affect results and how "familiar" response options might change things. However, I didn't really understand how this was theoretically relevant to the present investigation. ... it seems like this should either be cut or some more elaboration could be given.	This was to bring attention to the issue that, differences in the wording/connotations of instructions may influence results in non-trivial ways, and thus it might be worthwhile to study whether the effects were precisely dependent on certain instruction formats. We have now clarified the methodological importance of this point.	22
12	The introduction to Experiment 2A claims that not a lot of studies have conducted depth of processing manipulations with source memory,	We added this citation to the discussion of studies on LoP effects on source memory.	24

	but doesn't cite Glanzer, Hilford, and Kim (2004, Journal of Experimental Psychology: Learning, Memory, and Cognition). This paper conducts several manipulations on item recognition and source memory and their Experiment 3 found benefits of deep processing for both item recognition and source memory.		
--	--	--	--

Appendix B

Reviewer #1			
Point	Comment	Authors' Response	Page Reference
1	I downloaded the data from Dryad, but was disappointed not to see any statistical analysis code, eg in R – please provide that for a further review. Also, please do not share data files that require proprietary software like Excel – just save as a CSV file. Finally, there were several strange values in the Excel file like “#VALUE!” and “#N/A”. While “N/A” is fine to indicate missing data, please clarify the meaning of values prepended with a “#”.	The code has now been made available on Dryad. The data files have been converted to CSV format, with values such as “#VALUE!” and “#N/A” converted to blanks to reflect missing values.	n/a
2	The full paper PDF was not available until I asked the editor to re-upload. Once obtained, the PDF contained some formatting errors – eg on page 5, I see the phrase “Most dual-process accounts assume that familiarity is driven by a continuum of memory” repeated 7 times. I don't know whether I should ignore all these repetitions, or where there is a missing paragraph; another reason why I think a second, quick review is necessary.	We have now enclosed our own PDF version of the full paper, in order to ensure that there are minimal formatting errors.	n/a
3	It is interesting that there are many cases when the frequentist p-value and Bayes Factor diverge. I have experience of p-values just below .05 when BF10 is little more than 1, but I find it surprising when, e.g. on page 17, p is quite small (.0075) but BF10<1 (0.51). It might be worth double-checking the R code and priors used by BayesFactor package?	There were indeed errors in the R code for the BF10 values. We would like to apologise for this, and thank the reviewer for pointing them out. We have now recalculated and verified all the Bayes factors using JASP instead of the BayesFactor package, ensuring that (default) priors are consistently used across all analyses. However, the BF result for the interaction term in Expt 2a remains	17-19, 30-31, 36, 41-42, 45

		contradictory to its frequentist counterpart. We included a brief discussion of this pattern.	
4	I think “Following the a priori power and minimal sample size calculation from Experiment 3b...” should read “Following the a priori power and minimal sample size calculation from Experiment 3a...”	Corrected “Experiment 3b” to “Experiment 3a”.	43
5	Were random slopes allowed for source dimension, LoP and Experiment? Was the optimal random effects structure estimated first?	Originally in the previous draft, random slopes were not used for those variables, and neither were the optimal random effects structures estimated first. We are grateful for the reviewer’s suggestions, and we have now added the consideration of random slopes as part of the optimal random effects structure estimation process.	47-49, Tables 1&2
6	I think the discussion of the Lange et al study really helps, particularly in allaying the concern that the high correlation between priming and source memory is an artefact of sequential decisions on the same cue in the standard CID-R paradigm. While separating the two types of decision (into separate test phases) is important to rule out shared effects (eg of attention) during retrieval (a la Jacoby & Whitehouse), I’m not sure how “This suggests that any influence of correlations across items at encoding (e.g., as a result of attentional fluctuation) is minimal” – couldn’t fluctuations at encoding still produce correlation between tests on the same item, even if those tests occur in different sessions?	Yes, we agree that it could still be possible for correlated factors at encoding (e.g., attention) to produce correlations on priming and source memory at test. We have now clarified the wording of this section.	53
7	While the Dryad hyperlink at the start of the PDF is correct, I think	We have now amended the link under Data Availability to	55

	the link quoted in the final “Data Availability” section is incorrect.	the correct link at the beginning of the document.	
--	--	--	--

Reviewer #2			
Point	Comment	Authors’ Response	Page Reference
1.1	The Bayesian analyses included in the paper were not properly interpreted anywhere. It appears that the authors merely inserted the Bayes Factor (BF) results.	Interpretations of the Bayesian results have now been added in text. We thank the reviewer for kindly providing the relevant resources regarding the interpretations.	17-18, 20, 30, 32, 37, 42, 45
1.2	However, there are a number of places throughout the manuscript where the BF results are in direct contradiction to the frequentist results and this is not brought up or discussed. For instance:	See response to Reviewer #1, point 3.	17-19, 30-31, 36, 41-42, 45
3	There is some discussion about reversed association results within. It may help to just *briefly* cite and mention the following paper which did not replicate some of these important results: Haaf, J. M., Rhodes, S., Naveh-Benjamin, M. Sun, T., Snyder, K., & Rouder, J. N. (2020). Revisiting the remember-know task: Replications of Gardiner and Java (1990). Memory & Cognition .	We appreciate the reviewer’s suggestion of this recent paper, and have now included the citation.	6